# WINA: Weight Informed Neuron Activation for Accelerating Large Language Model Inference

**Sihan Chen**[2*]**, Dan Zhao**[1]**, Jongwoo Ko**[1]**, Colby Banbury**[1]**, Huiping Zhuang**[3]**,
Luming Liang**[1]**, Pashmina Cameron**[1]**, Tianyi Chen**[1†*]
[1]Microsoft, [2]Renmin University of China, [3]South China University of Technology
chensihan@ruc.edu.cn, Tianyi.Chen@microsoft.com

## Abstract

The ever-increasing computational demands of large language models (LLMs) make efficient inference a central challenge. While recent advances leverage specialized architectures or selective activation, they typically require (re)training or architectural modifications, limiting their broad applicability. Training-free sparse activation, in contrast, offers a plug-and-play pathway to efficiency; however, existing methods often rely solely on hidden state magnitudes, leading to significant approximation error and performance degradation. To address this, we introduce **WINA** (**W**eight-**I**nformed **N**euron **A**ctivation): a simple framework for training-free sparse activation that incorporates both hidden state magnitudes and weight matrix structure. By also leveraging the $\ell_2$-norm of the model's weight matrices, WINA yields a principled sparsification strategy with provably optimal approximation error bounds, offering better and tighter theoretical guarantees than prior state-of-the-art approaches. Overall, WINA also empirically outperforms many previous training-free methods across diverse LLM architectures and datasets: not only matching or exceeding their accuracy at comparable sparsity levels, but also sustaining performance better at more extreme sparsity levels. Together, these position WINA as a practical, theoretically grounded, and broadly deployable solution for efficient inference. Our code is available at github.com/microsoft/wina.

## 1 Introduction

While large language models (LLMs) have demonstrated impressive capabilities in a variety of applications such as text generation (Li et al., 2024b; Cheng et al., 2025), translation (Hendy et al., 2023; sea, 2025), understanding (Chang et al., 2024; Tschannen et al., 2025), and computer using agent (Xie et al., 2024; Hui et al., 2025). Their growing size and complexity often translate into a need for substantial computational resources, particularly during inference, making reducing inference costs without degrading output quality a key challenge.

One strategy is to only activate a sub-network (Jacobs et al., 1991) during inference through architectural changes such as Mixture of Experts (MoE), which has already seen adoption in LLMs like GPT4 (Achiam et al., 2023), Mistral (Jiang et al., 2023), etc., or through model distillation (Moslemi et al., 2024) where a smaller model is trained using knowledge distilled from a larger teacher model to route inference requests more efficiently. However, these approaches typically require a considerable amount of training which, in itself, can be computationally intense.

One alternative is training-free sparse activation (Liu et al., 2024a; Lee et al., 2024) which retains the original dense model but selectively omits weights/neurons at inference time. These methods avoid (re)training and can be applied to off-the-shelf models, leveraging criteria such as hidden-state magnitudes, weight importance, weight statistics, or additional validation data to determine which parts of the model to select to accelerate inference.

---

*Equal contributions. †Corresponding author.

Nonetheless, current training-free sparse-methods face several limitations. Most notably, they ignore the influence of weight matrices on error propagation: these approaches fail to account for how interactions between input elements and the weight matrix during forward propagation can affect model outputs, which lead to accumulating approximation errors in sparse activation.

**Contributions.** We propose WINA: a simple, easy-to-use, training-free framework that performs sparse activation based on both the magnitude of hidden states and the column-wise $\ell_2$-norm of the weight matrix. By combining activation strength with weight importance, our thresholds directly account for how much each activation can influence the next layer. This design provides theoretical guarantees on bounding the total approximation error in a way better than that of other approaches.

In contrast, methods like TEAL (Liu et al., 2024a) rely exclusively on the distribution of hidden-state magnitudes to decide which activations to keep. However, ignoring weight magnitudes in this way can discard highly influential activations while retaining many low-impact ones, leading to sub-optimal trade-offs between efficiency and output quality. These types of approaches overlook how the weight matrix directly influences input features during the forward pass; by focusing solely on activations, this omission can result in compounding approximation errors that ultimately skew the calculation of sparse activations and degrade the accuracy of the sparse model more than necessary. Our framework overcomes these limitations by integrating weight statistics into the selection process, achieving finer control over sparsity and tighter bounds on the resulting approximation error.

We evaluate WINA on several widely-used LLMs (ranging from 7B to 14B) across several popular benchmarks. Compared with existing training-free sparse activation methods like TEAL (Liu et al., 2024a), CATS (Lee et al., 2024), and R-Sparse (Zhang et al., 2025), WINA achieves superior model performance at

Table 1: Comparison between WINA and others.

|  | WINA | TEAL | CATS | R-Sparse |
|---|:---:|:---:|:---:|:---:|
| **Tight Approx Error** | ✓ | ✗ | ✗ | ✗ |
| **Layer-Agnostic Application**[†] | ✓ | ✓ | ✗ | ✓ |
| **Layer-Specific Sparsity** | ✓ | ✓ | ✗ | ✓ |

†: Some methods (i.e., CATS) are only adopted on specific types of layers.

comparable sparsity levels with significantly less performance degradation. We also establish tight theoretical error bounds for WINA, providing formal support for our experimental results to validate its effectiveness. In short, our contributions can be summarized as follows:

- **Weighted-Informed Activation.** We introduce a novel sparse activation method that jointly considers hidden state magnitudes and the column-wise $\ell_2$-norms of weight matrices. This allows for selecting neurons that are not only strongly activated but also those that have a larger influence on downstream layers, leading to a more informed construction of a sub-network during inference.

- **Theoretically Tighter Approximation Error.** We conduct a formal analysis to demonstrate that our weight-informed activation mechanism yields a lower expected output error compared to prior methods (e.g., TEAL) under mild assumptions.

- **Numerical Experiments.** We conduct extensive evaluations across multiple LLM families, including the Lllama series (Touvron et al., 2023), Phi-4 (Abdin et al., 2024), and Mistral (Jiang et al., 2023), and show that our method consistently achieves superior accuracy across a range of sparsity levels. In particular, WINA sustains higher performance as sparsity increases, highlighting its robustness and practical applicability across diverse tasks and model scale. Moreover, we also demonstrate WINA's compatibility with quantization, achieving promising results under both 4-bit and 8-bit settings, WINA's impact on models' long-context reasoning abilities, and its impact on social biases in models—aspects largely overlooked in prior works. Lastly, we provide a Triton kernel to benchmark WINA's competitive speed-ups against methods like TEAL.

The rest of our paper is organized as follows. We begin with related works in Section 2 and detail our methodology in Section 3. We present our results in Section 4 and conclude in Section 5.

## 2 RELATED WORK

**Sparse Activation.** Modern sparse activation approaches fall into two principal paradigms: training-based methods and training-free methods. Training-based methods typically employ a trainable router to learn to dynamically select activated experts for each token, with the Mixture-of-Experts (MoE) architecture (Jacobs et al., 1991) serving as the foundational framework.

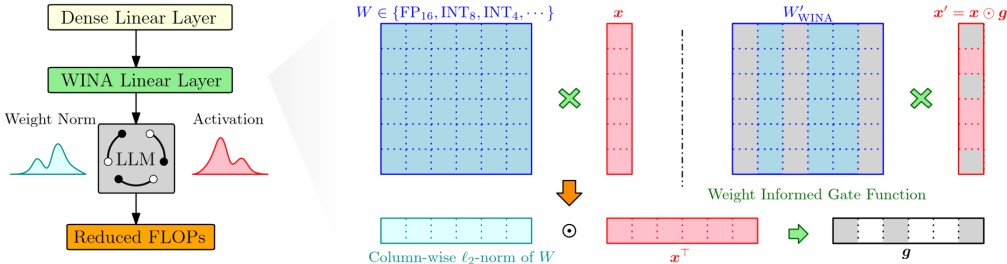

Figure 1: **Overview of WINA.** WINA performs training-free sparse activation by selecting the most influential input dimensions based on both activation magnitude and the column-wise $\ell_2$-norms of weight matrices. This joint criterion effectively extends to layers of varying precision, including quantized layers, ensuring accurate sub-network activation at each layer during inference.

This paradigm has been expanded through many iterations and variants. The sparsely-gated mixture of experts layer (Shazeer et al., 2017) integrates MoE into recurring neural networks (RNNs). Works like GShard (Lepikhin et al., 2020) and the Switch Transformer (Fedus et al., 2022) extend MoEs to the Transformer architecture (Raffel et al., 2020) while others combine several approaches, such as WideNet (Xue et al., 2022), reduces the size of the MoE model by initially compressing the model before transitioning into a MoE. Works like MoEBert (Zuo et al., 2022) decomposes the FFN layer of a pre-trained dense model into multiple experts based on importance-guided adaptation and then refines the model through distillation. LLM in Flash (Alizadeh et al., 2023) employs a low-rank predictor to determine which intermediate neurons are activated.

Training-free methods, in contrast, do not rely on a learnable router, instead using predefined criteria to perform sparse activation such as TEAL (Liu et al., 2024a), TDA (Ma et al., 2024), and SEAP (Liang et al., 2025). Methods (Han et al., 2015b) can utilize magnitude-based weight pruning or global activation pruning (Wen et al., 2016b) to apply a fixed sparsity pattern regardless of input. For instance, Q-Sparse (Wang et al., 2024) produces sparsity as a function of input magnitudes, achieving high sparsity with reasonable performance degradation. CATS (Lee et al., 2024) applies sparse activation on SwiGLU within gated MLP layers, achieving reasonable sparsity while maintaining performance. In contrast, TEAL (Liu et al., 2024a) extends magnitude-based activation sparsity to all layers, achieving high model-wide sparsity. However, current sparse activation methods tend to focus solely on selecting activation elements based on the magnitude of hidden states, which can result in suboptimal performance.

**Model Pruning.** Another line of related work is model pruning, which removes redundant neurons from deep neural networks (Han et al., 2015a; Frankle and Carbin, 2018; Frantar and Alistarh, 2023; Li et al., 2024a) to obtain compact yet high-performing sub-networks. While effective, pruning typically requires fine-tuning to recover accuracy (Lin et al., 2019; He et al., 2018; Wen et al., 2016a; Li et al., 2020; Zhuang et al., 2020; Chen et al., 2021b; 2024), introducing additional training overhead. Such retraining stages make pruning less practical for large foundation models.

## 3 METHODOLOGY

We now present WINA, a framework for sparse activation that preserves critical elements while zeroing out non-essential components in each layer's input. As illustrated in Figure 1, WINA jointly considers both the input activation and the associated weight matrix, rather than relying solely on activation magnitudes. During inference, it activates only the most influential neurons, effectively constructing a sparse sub-network that maintains the expressive power of the original model.

### 3.1 PROBLEM STATEMENT

**Problem.** Consider a deep neural network (DNN) $\mathcal{M}$ consisting of $L$ layers. We denote the weight matrix of the $l$-th layer as $W^{(l)} \in \mathbb{R}^{m_l \times n_l}$ and the corresponding input as an arbitrary tensor $X \in \mathbb{R}^{n_l \times s_l}$ for $l \in \{1, ..., L\}$, representing the full information content. Our goal is to identify a set of binary activation gates $\mathcal{G} = \{\boldsymbol{g}^{(1)}, \cdots, \boldsymbol{g}^{(L)}\}$, where each $\boldsymbol{g}^{(l)} \in \{0, 1\}^{n_l}$ is a binary vector, such that

the deviation between the model's original output and the gated output is minimized:

$$\underset{\boldsymbol{g}^{(1)},\cdots,\boldsymbol{g}^{(L)}}{\text{minimize}} \quad \|\mathcal{M}(X) - \mathcal{M}(X \mid \mathcal{G})\|_2. \tag{1}$$

Since obtaining the complete set of possible inputs $X$ is generally infeasible, we instead use a sampled subset $\tilde{X}$ for approximation. The activation gating operates in the input vector space to reduce output deviation. With this observation, we can reformulate the original problem into a more tractable per-layer version.

**Refined Problem.** Given a weight matrix $W \in \mathbb{R}^{m \times n}$ and input vector $\boldsymbol{x} \in \mathbb{R}^n$, the standard linear transformation is given by $\boldsymbol{y} \leftarrow W\boldsymbol{x}$. Our objective then becomes identifying an activation gate or mask $\boldsymbol{g} \in \{0,1\}^n$ such that the masked output $\boldsymbol{y_g} \leftarrow W(\boldsymbol{g} \odot \boldsymbol{x})$ approximates the original:

$$\underset{\boldsymbol{g} \in \{0,1\}^n}{\text{minimize}} \quad \|W\boldsymbol{x} - W(\boldsymbol{g} \odot \boldsymbol{x})\|_2. \tag{2}$$

## 3.2 WEIGHT INFORMED GATE FUNCTION

**Motivation.** Many current sparse activation methods (e.g., Q-sparse (Wang et al., 2024), CATS (Lee et al., 2024), TEAL (Liu et al., 2024a)) operate via a top-$K$ gating mechanism governed by the absolute values of the hidden states:

$$\boldsymbol{g}_i = \begin{cases} 1 & \text{if } |\boldsymbol{x}_i| \text{ is among the top-}K \text{ values in } |\boldsymbol{x}|, \\ 0 & \text{otherwise.} \end{cases} \tag{3}$$

However, this approach ignores the critical role that weight matrices themselves play: more specifically, it ignores how each element of the preceding input interacts with the weight matrix $W$ during the forward pass. This mismatch is what motivates us to propose WINA, a method that jointly considers both inputs and weight matrices to minimize the approximation error for better performance.

In WINA, we construct binary activation gates by selecting the top-$K$ components via:

$$[\boldsymbol{g}_{\text{WINA}}]_i = \begin{cases} 1 & \text{if } |\boldsymbol{x}_i \boldsymbol{c}_i| \text{ is among the top-}K \text{ values in } |\boldsymbol{x} \odot \boldsymbol{c}|, \\ 0 & \text{otherwise,} \end{cases} \tag{4}$$

where $\boldsymbol{c} \in \mathbb{R}^n$ represents the column-wise $\ell_2$-norm of $W$ and $\odot$ denotes the Hadamard or element-wise product. A smaller $K$ results in more deactivated neurons, thereby saving more FLOPs while potentially sacrificing performance. The choice of $K$ is flexible and adaptable, ranging from a coarse-grained universal criterion where a shared $K$ is applied across all layers to a fine-grained layer-specific strategy that assigns $K$ individually to better minimize performance degradation. In the meantime, WINA is architecture agnostic, applicable across different layers, such as attention layers, multi-layer perceptions (MLPs), or residual connections, etc.

## 3.3 THEORETICALLY OPTIMAL APPROXIMATION ERROR

WINA also comes with theoretical guarantees, establishing a tighter bound on the approximation error than prior approaches under relatively mild assumptions. We first show that for a single linear layer network, WINA yields an optimal solution to the target problem (Eqn. 2).

**Lemma 3.1** (Optimal approximation error over single linear layer). *Let $\boldsymbol{x} \in \mathbb{R}^n$ be an input vector and $W \in \mathbb{R}^{m \times n}$ be a matrix satisfying column-wise orthogonality: $W^\top W$ is a diagonal matrix. For any target density level $K \in \mathbb{N}^+$ satisfying $K < n$, the deviation between the original network output and the gated output via WINA is optimal. Formally,*

$$\boldsymbol{g}_{\text{WINA}}(\boldsymbol{x}) = \underset{\boldsymbol{g} \in \{0,1\}^n}{\arg\min} \quad \|W\boldsymbol{x} - W(\boldsymbol{g} \odot \boldsymbol{x})\|_2,$$

*where $\boldsymbol{x}$ is the input, and $\boldsymbol{g}_{\text{WINA}}(\boldsymbol{x})$ is the gating function of WINA retaining the $K$ elements activated with the largest $|x_i \cdot \|W_{\cdot,i}\|_2|$ for $i \in \{1, \cdots, n\}$.*

**Proof.** See Appendix A.7.

Building upon our single linear-layer Lemma 3.1, we now extend it to deep $L$ linear layer networks $\mathcal{M}$ and present that WINA could tighten the gated error upper bound.

**Theorem 3.2** (WINA minimizes a provable upper bound on output deviation). *Consider an L-layer linear network $\mathcal{M}(\boldsymbol{x}) = W^{(L)}W^{(L-1)}...W^{(1)}\boldsymbol{x}$, where for each $\ell \geq 2$ the weight matrix satisfies $W^{(\ell)\top}W^{(\ell)} = D^{(\ell)}$ with $D^{(\ell)}$ diagonal (column-orthogonality). Let $\mathcal{G} = \{\boldsymbol{g}^{(1)}, \ldots, \boldsymbol{g}^{(L)}\}$ be gating variables with $\boldsymbol{g}^{(\ell)} \in \{0,1\}^{d_\ell}$ and $M^{(\ell)} = \mathrm{diag}(\boldsymbol{g}^{(\ell)})$. Let $\mathcal{M}(\boldsymbol{x} \mid \mathcal{G})$ denote the gated network output. Define the output deviation $E(\boldsymbol{x};\mathcal{G}) := \|\mathcal{M}(\boldsymbol{x}) - \mathcal{M}(\boldsymbol{x} \mid \mathcal{G})\|_2^2$. Then there exists a separable upper bound $E(\boldsymbol{x};\mathcal{G}) \leq \mathcal{U}(\boldsymbol{x};\mathcal{G})$. Moreover, minimizing $\mathcal{U}(\boldsymbol{x};\mathcal{G})$ reduces to selecting, at each layer, the $k$ largest coordinates weighted by squared column norms. Therefore WINA satisfies*

$$\mathcal{G}_{\mathrm{WINA}} = \arg\min_{\mathcal{G}} \mathcal{U}(\boldsymbol{x};\mathcal{G}).$$

**Proof.** See Appendix A.8 for details.

**Remarks on Column-Wise Orthogonality.** Our analysis relies on the column-wise orthogonality of the relevant weight matrices. We applied an efficient one-off offline tensor transformation from (Ashkboos et al., 2024a) to enforce column orthogonality. This pre-processing step is lightweight, does not change the functional capacity of the model, and enables our theoretical guarantees to translate effectively to practical settings. We leave details to Appendix A.2.

## 3.4 THEORETICAL VALIDATION WITH SYNTHETIC EXPERIMENTS

To validate our theoretical analysis, we first conduct controlled synthetic experiments on randomly initialized networks under the assumptions of Lemma 3.1 and Theorem 3.2.

We initialize input vectors and weight matrices using Kaiming initialization with the SiLU activation function, and enforce column-wise orthogonality of weight matrices via the tensor transformation described in Appendix A.2. We then compare the dense network output against sparsified outputs generated by CATS, TEAL, and R-Sparse. Since CATS and TEAL share the same underlying mechanism and differ only in their sparsity distribution, we group them together as a baseline.

Performance is quantified by the $\ell_2$ deviation between the dense output and the sparsified outputs across varying sparsity ratios. Each experiment is repeated with 20 random seeds. we report averaged results associated with detailed error bar analysis in Table 2. WINA consistently achieves lower approx-

Table 2: Approximation errors of different methods over randomly initialized networks and sparsity levels. Lower is better.

| Theory | Method | 25% | 40% | 50% | 65% |
|---|---|---|---|---|---|
| Lemma 3.1 | CATS/TEAL | 1.68 ± 0.14 | 3.41 ± 0.20 | 4.86 ± 0.23 | 7.55 ± 0.32 |
| | R-Sparse | 1.72 ± 0.13 | 3.48 ± 0.20 | 5.01 ± 0.30 | 7.75 ± 0.35 |
| | **WINA** | **0.70 ± 0.05** | **1.73 ± 0.09** | **2.70 ± 0.13** | **4.75 ± 0.15** |
| Theorem 3.2 | CATS/TEAL | 0.73 ± 0.04 | 1.44 ± 0.06 | 2.04 ± 0.11 | 3.02 ± 0.18 |
| | R-Sparse | 0.77 ± 0.04 | 1.51 ± 0.07 | 2.11 ± 0.11 | 3.13 ± 0.16 |
| | **WINA** | **0.38 ± 0.02** | **0.76 ± 0.04** | **1.09 ± 0.06** | **1.76 ± 0.08** |

imation error than competing methods across all sparsity levels and theoretical settings. Notably, WINA reduces error about **50%** compared to others, aligning with our theoretical guarantees. Building on this, we turn next to evaluating WINA on LLMs in more realistic settings.

# 4 EXPERIMENTS

## 4.1 SETUP

**Models.** To demonstrate WINA's effectiveness across different model families and sizes, we provide our results on four commonly used LLMs: Llama-2-7B (Touvron et al., 2023), Llama-3-8B (Dubey et al., 2024), Mistral-7B (Jiang et al., 2023), and Phi-4-14B (Abdin et al., 2024).

**Evaluation.** We use the lm-evaluation-harness pipeline (Gao et al., 2023) to assess WINA across a diverse suite of tasks. In addition to the commonsense reasoning examined in works like R-Sparse (Zhang et al., 2025), we also consider general reasoning, mathematics, and code generation. For commonsense reasoning, we evaluate on PIQA (Bisk et al., 2020), WinoGrande (Sakaguchi et al., 2019), HellaSwag (Zellers et al., 2019), BoolQ (Clark et al., 2019), Arc Challenge and Arc Easy (Clark et al., 2018), SciQ (Johannes Welbl, 2017), and OpenBookQA (Mihaylov et al., 2018), which measure capabilities such as applying everyday knowledge and resolving coreferences to and

answer questions. For general reasoning and knowledge, we consider MMLU (Hendrycks et al., 2020), which spans domains across STEM, humanities, and social sciences, providing a comprehensive test of broad reasoning and knowledge transfer. For math, we adopt GSM8K (Cobbe et al., 2021), which examines multi-step arithmetic problem solving. For code generation, we use HumanEval (Chen et al., 2021a) which requires synthesizing executable Python code. We note that WINA has a closed-form mechanism characterized in Eqn. (4), making it deterministic given an input. Therefore, there are no error bars to report for WINA's performance.

**Baselines.** We compare WINA with three recent, representative sparse activation methods: CATS (Lee et al., 2024), R-Sparse (Zhang et al., 2025), and TEAL (Liu et al., 2024a). We assign layer-specific sparsity ratios instead of a uniform sparsity across the model so that, given a global sparsity target, we leverage the greedy algorithm as proposed in TEAL (Liu et al., 2024a) to iteratively configure per-layer sparsity levels so that the aggregate sparsity meets the global budget.

## 4.2 RESULTS ON COMMONSENSE REASONING

Following the experimental setup of works like R-Sparse (Zhang et al., 2025), we provide an empirical comparison of WINA against various baselines (e.g., CATS, R-Sparse, and TEAL) across different sparsity levels, ranging from 25% to 65% on commonsense reasoning tasks to demonstrate effectiveness across various experimental settings. Figure 2 provides an comparative overview of the performance-sparsity trade-offs across the different methods.

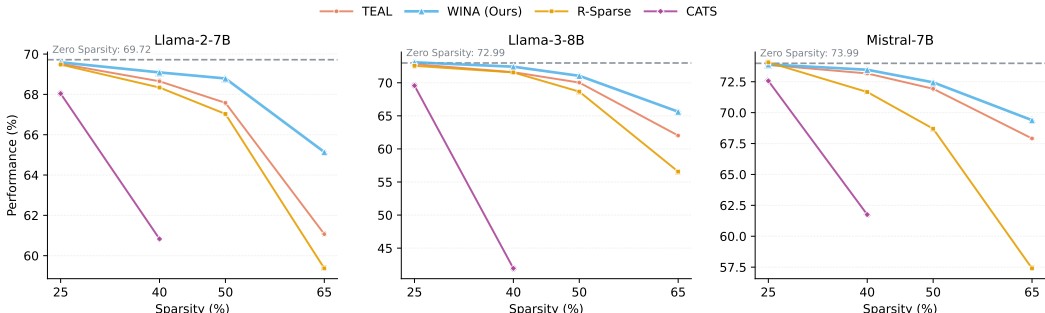

Figure 2: **Performance-Sparsity Frontier.** Mapping out average performance at each sparsity level for various techniques. WINA outperforms other methods (TEAL, R-Sparse, CATS) across sparsity levels with the performance gap increasing as sparsity increases. We note that CATS is unable to achieve higher model sparsity due to its applicability to only certain layers.

Table 3: Results over Llama-2-7B on commensense reasoning.

| Sparsity | Method | PiQA | Arc-C | WinoGrande | HellaSwag | SciQ | OBQA | BoolQ | Arc-E | Avg |
|---|---|---|---|---|---|---|---|---|---|---|
| 0% | Baseline (full model) | 79.05 | 46.25 | 68.90 | 76.00 | 91.00 | 44.20 | 77.77 | 74.58 | 69.72 |
| 25% | CATS[†] | 77.20 | 44.20 | 67.48 | 75.75 | 90.10 | 43.40 | 74.31 | 71.89 | 68.04 |
| | R-Sparse | 78.89 | 45.99 | 68.35 | 75.60 | 91.70 | 43.00 | 77.71 | 74.58 | 69.48 |
| | TEAL | 78.73 | 45.99 | 68.90 | 75.98 | 91.00 | 44.20 | 77.09 | 74.24 | 69.52 |
| | **WINA** | 78.40 | 46.16 | 69.38 | 75.93 | 90.90 | 44.00 | 77.03 | 74.92 | **69.59** |
| 40% | CATS[†] | 74.70 | 36.86 | 61.88 | 69.25 | 79.90 | 39.00 | 66.06 | 58.96 | 60.83 |
| | R-Sparse | 78.18 | 43.60 | 67.09 | 74.57 | 91.40 | 42.80 | 75.72 | 73.32 | 68.34 |
| | TEAL | 78.07 | 45.22 | 66.85 | 75.31 | 91.40 | 43.20 | 76.48 | 72.69 | 68.65 |
| | **WINA** | 78.40 | 45.31 | 68.59 | 75.48 | 91.50 | 42.40 | 76.79 | 74.28 | **69.09** |
| 50% | R-Sparse | 76.93 | 42.32 | 65.82 | 72.25 | 91.60 | 41.00 | 74.46 | 71.84 | 67.03 |
| | TEAL | 77.09 | 42.92 | 68.03 | 73.54 | 90.20 | 43.20 | 74.07 | 71.55 | 67.58 |
| | **WINA** | 77.80 | 44.71 | 68.51 | 74.43 | 91.00 | 44.60 | 75.84 | 73.44 | **68.79** |
| 65% | R-Sparse | 71.93 | 34.22 | 59.04 | 59.63 | 87.50 | 34.20 | 66.91 | 61.53 | 59.37 |
| | TEAL | 73.61 | 36.09 | 62.12 | 63.58 | 84.00 | 38.00 | 69.24 | 61.95 | 61.07 |
| | **WINA** | 75.35 | 39.68 | 65.82 | 69.03 | 89.50 | 40.40 | 72.78 | 68.60 | **65.14** |

† CATS is unable to reach 50% or 65% sparsity since it only achieves sparse activations over MLP layers.

Table 4: Results over Llama-3-8B on commonsense reasoning.

| Sparsity | Method | PiQA | Arc-C | WinoGrande | HellaSwag | SciQ | OBQA | BoolQ | Arc-E | Avg |
|---|---|---|---|---|---|---|---|---|---|---|
| 0% | Baseline (full model) | 80.79 | 53.33 | 72.61 | 79.17 | 93.90 | 45.00 | 81.38 | 77.74 | 72.99 |
| 25% | CATS† | 78.62 | 48.04 | 70.64 | 76.32 | 91.90 | 41.80 | 78.13 | 71.09 | 69.57 |
| | R-Sparse | 79.82 | 52.05 | 72.38 | 78.69 | 93.50 | 44.40 | 80.92 | 78.75 | 72.56 |
| | TEAL | 80.20 | 53.16 | 73.32 | 78.85 | 94.10 | 45.20 | 80.83 | 76.89 | 72.82 |
| | **WINA** | 80.41 | 52.82 | 73.80 | 78.99 | 94.00 | 44.60 | 82.05 | 78.03 | **73.09** |
| 40% | CATS† | 59.96 | 27.82 | 51.30 | 40.18 | 46.10 | 29.80 | 42.26 | 38.09 | 41.94 |
| | R-Sparse | 79.05 | 50.26 | 72.14 | 76.91 | 94.10 | 43.00 | 79.14 | 77.86 | 71.56 |
| | TEAL | 79.00 | 48.98 | 71.82 | 77.45 | 93.30 | 45.00 | 80.03 | 77.19 | 71.60 |
| | **WINA** | 79.87 | 50.68 | 72.30 | 77.91 | 93.90 | 45.00 | 82.23 | 77.57 | **72.43** |
| 50% | R-Sparse | 76.22 | 45.73 | 66.61 | 73.22 | 93.80 | 42.20 | 76.70 | 74.83 | 68.66 |
| | TEAL | 78.29 | 48.12 | 70.09 | 74.83 | 93.70 | 42.60 | 78.23 | 74.41 | 70.03 |
| | **WINA** | 79.16 | 48.81 | 70.64 | 76.44 | 93.50 | 43.60 | 81.25 | 75.00 | **71.05** |
| 65% | R-Sparse | 68.50 | 33.36 | 57.38 | 51.48 | 86.00 | 31.80 | 65.23 | 58.80 | 56.57 |
| | TEAL | 73.34 | 37.37 | 63.46 | 61.76 | 88.90 | 37.00 | 69.85 | 64.48 | 62.02 |
| | **WINA** | 74.65 | 41.98 | 64.48 | 67.89 | 90.70 | 41.60 | 76.73 | 67.00 | **65.63** |

† CATS is unable to reach 50% or 65% sparsity since it only achieves sparse activations over MLP layers.

**Llama-2-7B.** On Llama-2-7B, WINA shows strong performance under various sparsity constraints. As shown in Table 3, WINA achieves the highest average accuracy at 25% sparsity, outperforming CATS, R-Sparse and TEAL. While performance naturally degrades at the more extreme 65% sparsity level, WINA still offers the best accuracy, scoring **+5.77%** higher than R-Sparse and **+4.07%** higher than TEAL, suggesting its robustness under aggressive pruning.

**Llama-3-8B.** Across all sparsity levels, WINA (Table 4) consistently outperforms or matches the best baseline methods. At 25% sparsity, WINA achieves the highest average score of 73.09%, surpassing the full baseline model (72.99%) and other sparse methods. This trend continues at higher sparsity levels: WINA maintains strong performance at 40% and 50% sparsity with average scores of 72.43% and 71.05%, respectively. Even at 65% sparsity, WINA remains competitive while other methods experience significant degradation, on average scoring **+3.61%** higher than TEAL and **+9.06%** higher than R-Sparse.

Table 5: Results over Mistral-7B on commensense reasoning.

| Sparsity | Method | PiQA | Arc-C | WinoGrande | HellaSwag | SciQ | OBQA | BoolQ | Arc-E | Avg |
|---|---|---|---|---|---|---|---|---|---|---|
| 0% | Baseline (full model) | 82.05 | 54.01 | 73.88 | 81.06 | 93.90 | 43.80 | 83.61 | 79.59 | 73.99 |
| 25% | CATS† | 81.01 | 52.22 | 72.38 | 80.86 | 92.00 | 42.40 | 82.60 | 77.06 | 72.57 |
| | R-Sparse | 81.61 | 53.41 | 74.19 | 81.06 | 94.10 | 45.60 | 83.73 | 78.83 | **74.07** |
| | TEAL | 82.21 | 53.75 | 73.48 | 80.93 | 94.10 | 44.00 | 83.52 | 78.87 | 73.86 |
| | **WINA** | 82.10 | 53.24 | 74.35 | 80.88 | 93.90 | 43.60 | 83.30 | 79.80 | 73.90 |
| 40% | CATS† | 75.95 | 41.13 | 63.61 | 69.27 | 78.90 | 36.20 | 68.07 | 60.86 | 61.75 |
| | R-Sparse | 81.77 | 51.62 | 70.09 | 77.56 | 91.50 | 39.20 | 83.09 | 78.54 | 71.67 |
| | TEAL | 81.56 | 52.30 | 73.40 | 80.03 | 93.80 | 43.00 | 83.21 | 78.07 | 73.17 |
| | **WINA** | 82.05 | 53.33 | 72.38 | 80.41 | 94.30 | 43.60 | 82.87 | 78.83 | **73.47** |
| 50% | R-Sparse | 81.07 | 49.74 | 66.22 | 69.39 | 89.50 | 34.20 | 82.14 | 77.36 | 68.70 |
| | TEAL | 79.76 | 49.57 | 70.80 | 78.84 | 93.90 | 43.00 | 82.75 | 76.81 | 71.93 |
| | **WINA** | 81.34 | 52.30 | 70.88 | 79.60 | 94.50 | 41.80 | 81.59 | 77.57 | **72.45** |
| 65% | R-Sparse | 63.60 | 36.52 | 63.77 | 59.30 | 88.00 | 30.20 | 74.16 | 43.64 | 57.40 |
| | TEAL | 77.97 | 42.49 | 66.22 | 72.26 | 92.20 | 40.40 | 79.27 | 72.47 | 67.91 |
| | **WINA** | 78.24 | 48.12 | 66.38 | 75.11 | 92.90 | 42.00 | 77.65 | 74.75 | **69.39** |

† CATS is unable to reach 50% or 65% sparsity since it only achieves sparse activations over MLP layers.

**Mistral-7B.** Overall, WINA matches or outperforms or matches the best performing methods across different sparsity levels (Table 5). At moderate sparsity (25-40%), WINA maintains performance nearly identical to the full model baseline, showing strong robustness to pruning. WINA's performance becomes most evident under extreme sparsity: at 65% sparsity, WINA achieves an average score of 69.39%, which is **+9.48%** higher than R-Sparse (59.91%) and **+1.48%** higher than TEAL (67.91%), demonstrating that WINA is more effective under aggressive compression.

### 4.3 ADDITIONAL ANALYSIS AND ABLATIONS

**Additional Benchmarks and Architecture.** We conduct additional experiments on several advanced tasks, including general reasoning (MMLU), mathematics (GSM8K), and coding (HumanEval). Additionally, we use Phi-4-14B to test the adaptability of WINA to different model architectures. To focus our analysis and reduce clutter, we compare against TEAL, the second best performing method in Section 4.2 as an upper-bound. We detail our results in Table 6.

We observe that WINA consistently delivers better performance than TEAL across all sparsity levels and tasks, highlighting its robustness on diverse applications, ranging from commonsense reasoning and subject knowledge to mathematical reasoning and code generation. Similar to our findings from Section 4.2, WINA's performance advantage becomes even more pronounced at higher sparsity levels, demonstrating its superior scalability. At 65% sparsity, the improvements over TEAL are considerable: **+1.65%** for Commonsense Reasoning, **+4.88%** for MMLU, **+2.73%** for GSM8K, and **+9.14%** for HumanEval. Notably, on HumanEval, WINA even surpasses the baseline, a phenomenon often observed in pruning literature (e.g., see Section 2) as its sparse activation may also be implicitly regularizing or suppressing harmful neurons, thereby enhancing performance.

Table 6: Results over Phi-4-14B over additional benchmarks.

| Sparsity | Method | Commonsense (avg) | MMLU | GSM8K | HumanEval |
|---|---|---|---|---|---|
| 0% | Baseline (full model) | 74.38 | 77.06 | 90.22 | 50.61 |
| 25% | TEAL | 74.12 | 76.63 | 89.84 | 46.95 |
| | **WINA** | **74.25** | **76.60** | **90.22** | **50.00** |
| 40% | TEAL | 73.49 | 75.10 | 88.02 | 45.73 |
| | **WINA** | **73.76** | **76.44** | **90.67** | **53.00** |
| 50% | TEAL | 72.73 | 73.52 | 86.13 | 41.46 |
| | **WINA** | **73.51** | **75.83** | **87.57** | **51.83** |
| 65% | TEAL | 69.93 | 65.17 | 74.37 | 32.32 |
| | **WINA** | **71.58** | **70.05** | **77.10** | **41.46** |

**Computational Savings.** In addition to performance gains, WINA also yields computational savings; as shown in Figure 3, WINA reduces the overall (G)FLOPs by up to 63.7% on Llama-2-7B and Mistral-7B, 60.4% on Llama-3-8B, and 62.7% on Phi-4-14B at 65% sparsity, potentially translating to faster inference and lower computational costs under tight resource constraints.

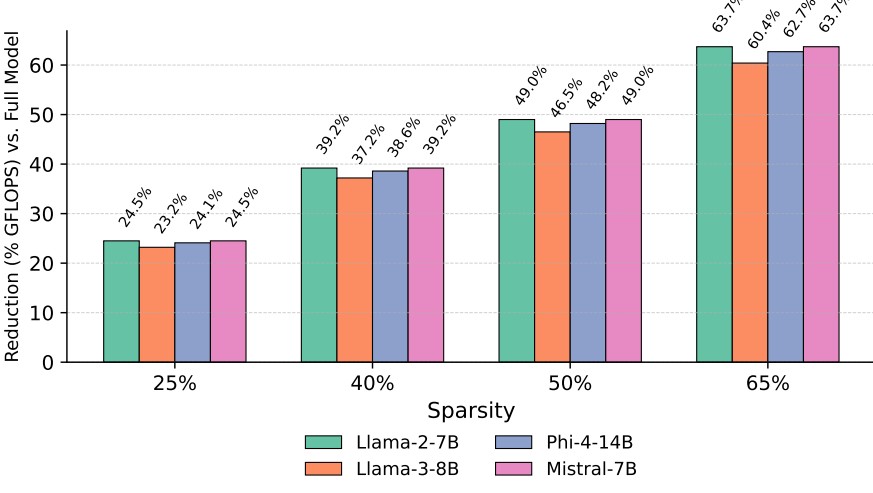

Figure 3: **Computational Savings from WINA.** Percentages indicate the reduction in GFLOPs at the specified sparsity level vs. the full dense model (higher is better).

**Compatibility with Quantization.** In resource constrained environments, LLMs are typically deployed in quantized form or lower precision (Ding et al., 2023). Although prior works rarely evaluate the robustness of their methods under quantization, we show that WINA is naturally compatible and apply it onto quantized versions of Llama-2-7B to assess performance on diverse commonsense reasoning tasks. As shown in Table 7, WINA incurs only minimal degradation, consistently outperforming TEAL; similar to before, the performance gap between WINA and TEAL increases as sparsity increases. Most notably, at 65%

Table 7: Llama-2-7B results across sparsity and quantization levels on commonsense reasoning.

| Sparsity | Method | $FP_{16}$ | $INT_8$ | $INT_4$ |
|---|---|---|---|---|
| 0% | Baseline (full model) | 69.72 | 69.39 | 68.06 |
| 25% | TEAL | 69.52 | 68.09 | 67.93 |
| | **WINA** | **69.59** | **68.39** | **68.06** |
| 40% | TEAL | 68.65 | 67.66 | 67.39 |
| | **WINA** | **69.09** | **68.16** | **67.52** |
| 50% | TEAL | 67.58 | 66.41 | 66.00 |
| | **WINA** | **68.79** | **67.28** | **66.50** |
| 65% | TEAL | 61.07 | 60.38 | 59.99 |
| | **WINA** | **65.14** | **64.24** | **63.67** |

sparsity, WINA outperforms TEAL by **+4.07%** on 16-bit, **+3.86%** on 8-bit, and **+3.68%** on 4-bit quantized models. We leave the treatment of sparsification-aware quantization (e.g., joint sparse-activation-quantization (Lin et al., 2024)), to future work.

**Ablative Study on Orthogonalization.** To control for potential confounding effects from the transformation process, we introduce an additional baseline, TEAL-Transform, where TEAL is also applied to the transformed model, retaining the $K$ elements with the largest absolute values $|x|$. Table 8 shows the average performance over commonsense datasets from our main experiments. Consistent with earlier results, WINA still outperforms in most cases with the gap widening as sparsity increases,

Table 8: Ablating the effects of orthogonalization.

| Method | Sparsity | Llama-2-7B | Llama-3-8B |
|---|---|---|---|
| 25% | TEAL (transform) | **69.70** | 72.91 |
| | **WINA** | 69.59 | **73.09** |
| 40% | TEAL (transform) | **69.11** | 72.05 |
| | **WINA** | 69.09 | **72.43** |
| 50% | TEAL (transform) | 67.97 | 70.36 |
| | **WINA** | **68.79** | **71.05** |
| 65% | TEAL (transform) | 63.05 | 63.52 |
| | **WINA** | **65.14** | **65.63** |

demonstrating that gains arise primarily from WINA's gating function.

**Social Bias.** To account for the effect of model compression on potential social/model biases, following (Gonçalves and Strubell, 2023), we evaluate our sparsified models on CrowS-Pairs (Table 13). Across all models and sparsity levels, we observe no systematic increase in bias, suggesting WINA does not exacerbate bias and can modestly mitigate it. We leave full details to Appendix A.6.

### 4.4 HARDWARE ACCELERATION

To achieve realistic speed-ups, we develop a dedicated sparse GEMV kernel for WINA using Triton (Tillet et al., 2019). This kernel takes the input tensor $x$, matrix $W$, pre-computed column-wise norm $c$ of $W$, and a desired sparsity level to return an output $x \odot g$. WINA gating $g$ is computed via Eq. (4), selecting a subset of columns of $W$ before the input $x$ is fed into the matrix multiply. The weight norms $c$ are pre-computed offline once during model loading. Compared to TEAL's sparse kernel, WINA only introduces an additional element-wise product $x \odot c$, whose cost is negligible.

In particular, let $d$ be the hidden dimension, $B$ be the batch size, and $T$ be the sequence length so that a standard linear layer incurs $\mathcal{O}(BTd^2)$ FLOPs during the forward pass while WINA's gating mechanism requires $\mathcal{O}(BTd)$ FLOPs. Comparing the two, their ratio is on order of $\mathcal{O}(BTd)/\mathcal{O}(BTd^2) = \mathcal{O}(1/d)$. Since $d$ is typically large in modern LLMs, (e.g, 2048, 4096 or larger), WINA consumes only $\ll 0.1\%$ additional overhead, which is negligible compared to the acceleration and accuracy improvements that its sparsification provides.

To empirically verify the negligible runtime overhead, we benchmark our Triton WINA kernel across commonly used matrix shapes in modern LLMs and batch sizes $1, 64, 256$. To avoid hardware-specific bias, we profile on three GPU architectures: A800, A100, and the RTX PRO 6000 Blackwell. Results in Figure 4 show that WINA matches TEAL's speed/latency almost identically. This phenomenon is consistent across all tested hardware platforms and batch sizes. Due to space considerations, we leave results on A800 and additional details to Appendix A.10. Additionally, we investigate how the speed-up ratio varies with the batch size ( Appendix A.4). As batch size increases

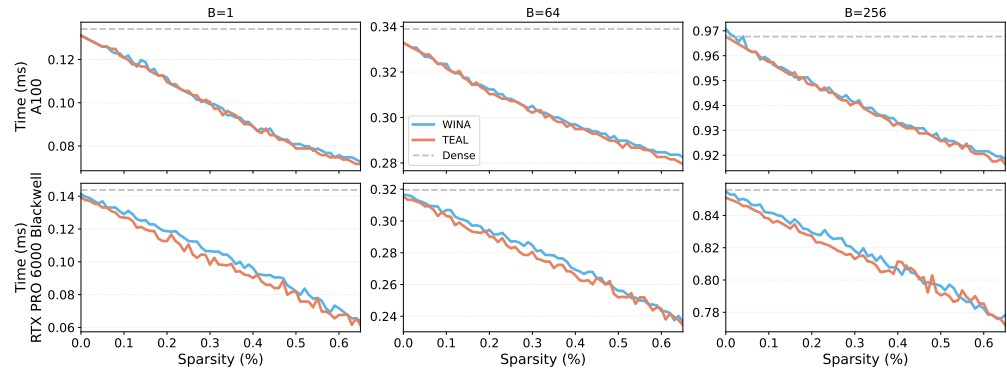

Figure 4: Sparsity vs. latency across different batch sizes $B \in \{1, 64, 256\}$ for GEMV (generalized matrix-vector multiplication) of sizes $5120 \times 1$ and $5120 \times 17920$. WINA's Triton kernel performance consistently matches that of TEAL across different GPU architectures (A100 top, RTX PRO 6000 Blackwell bottom), achieving similar speedups across our sparsity levels and as sparsity increases.

and GPU memory bandwidth becomes saturated, the achievable speed-up gradually decreases. Nevertheless, degradation is mild and WINA still delivers substantial acceleration, achieving approximately up to 42% speed-up on A100 and up to 57% RTX PRO 6000 Blackwell across different batch sizes, similar to TEAL.

### 4.5 PERFORMANCE ON LONG CONTEXT TASKS

We further evaluate the long-context reasoning ability of models under WINA on LongBench (Bai et al., 2024), a multi-task benchmark with realistic long-context scenarios involving code completion, summarization, as well as single and multi-document QA. LongBench contains 4,750 examples with average task length ranging from 5-15K. We evaluated Llama-2-7B, Llama-3-8B, and Phi-4-14B, which support context windows of 4K, 8K, and 16K tokens, respectively. Table 9 shows results for Llama-2-7B at 4K where, across all settings, WINA consistently outperforms TEAL even in more demanding scenarios. We leave additional results to Appendix A.5.

Table 9: Performance on LongBench (Llama-2-7B-4K).

| Sparsity | Method | Code Completion | Few-shot Learning | Summari-zation | Multi-Document QA | Single-Document QA | Synthetic Tasks | Overall |
|---|---|---|---|---|---|---|---|---|
| 0 | Baseline | 62.28 | 52.09 | 11.63 | 6.73 | 13.00 | 5.56 | 22.62 |
| 0.25 | TEAL | 62.14 | 51.80 | 11.83 | 7.24 | 12.87 | 5.06 | 22.59 |
| | **WINA** | 62.54 | 52.46 | 12.48 | 7.07 | 12.95 | 5.26 | **22.89** |
| 0.4 | TEAL | 60.28 | 52.23 | 12.55 | 8.12 | 13.40 | 4.54 | 22.83 |
| | **WINA** | 61.52 | 51.76 | 13.81 | 7.63 | 13.64 | 4.97 | **23.11** |
| 0.5 | TEAL | 58.41 | 50.10 | 12.97 | 8.39 | 12.80 | 4.28 | 22.23 |
| | **WINA** | 61.15 | 52.16 | 14.62 | 7.41 | 12.09 | 3.13 | **22.71** |
| 0.65 | TEAL | 44.20 | 44.15 | 11.16 | 7.09 | 8.17 | 1.62 | 17.88 |
| | **WINA** | 55.03 | 48.93 | 8.98 | 6.42 | 8.60 | 2.85 | **19.54** |

## 5 CONCLUSION

WINA offers not only a simple yet highly effective training-free sparse-activation for accelerating LLMs but also solid theoretical guarantees on its approximation error. Our results show that it consistently outperforms strong baselines across many benchmarks, while maintaining robustness even under aggressive 4-bit and 8-bit quantization. These results highlight both the practicality and the plug-and-play usability of WINA, making it ideal for sparse-activation-accelerated inference.

## ETHICS STATEMENT

Our work introduces a training-free sparse activation framework designed to improve the efficiency of large language model inference. The primary ethical consideration is ensuring that such efficiency gains are not misused to scale potentially harmful or biased models more widely without proper safeguards. We emphasize that WINA is a general-purpose acceleration method and does not alter the underlying datasets or model outputs. As such, any societal risks or biases present in the base models remain unchanged, and practitioners must remain vigilant about responsible deployment. Additionally, by lowering computational costs, our approach may promote accessibility of research to institutions with limited resources, contributing to broader inclusivity in AI research.

## REPRODUCIBILITY STATEMENT

We have taken steps to ensure the reproducibility of our results. All experiments were conducted on publicly available benchmark datasets. We used standard open-source evaluation pipelines such as lm-evaluation-harness for benchmarking. Detailed experimental settings are reported in Section 4, including models (e.g., Llama-2, Llama-3, Mistral-7B, and Phi-4), sparsity configurations, and computational resources. Comparisons were made against established baselines under controlled sparsity levels, and we provide theoretical analyses and ablation studies to support our claims. The total run time of our experiments were run using one A100 80GB GPU for several days along with a few hours on a RTX PRO 6000 Blackwell and A800 for minor benchmarking. Our source code is made anonymously available at URL.

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

# A  Appendix

## A.1  The Use of Large Language Models (LLMs)

The use of LLMs in this paper was restricted to limited editing supports, such as detecting and correcting grammar errors, typos, rephrasing sentences.

## A.2  Orthogonal Tensor Transformation

To enforce orthogonality in DNNs, there are two families of approaches. The first one is to augment an orthogonality regularizer $\|W^\top W - I\|$ into the objective function (Xie et al., 2017; Huang et al., 2020). This line of approach requires some model training. Another family of approaches uses matrix transformation or rotation to enforce orthogonality. In particular, SliceGPT (Ashkboos et al., 2024a) propose a transformation to bring orthogonality while preserve the numerical equivalence or computation invariance. Similar transformation is also widely used in efficient-AI literatures (Ashkboos et al., 2024b; Chee et al., 2023; Liu et al., 2024b; Hu et al., 2025). In this work, we adopt the same transformation from (Ashkboos et al., 2024a) to produce computational invariant LLMs with enforced orthogonality to maximize the performance gain of WINA.

**Overview.**  Given a weight matrix $W$, we can enforce column-wise orthogonality by multiplying $W$ from the right by an orthogonal matrix $Q$ such that the product $WQ$ has orthogonal columns. Specifically, we perform Singular Value Decomposition (SVD) on $W$, $W = U\Sigma V^\top$, where $U$ and $V$ are orthogonal matrices, and $\Sigma$ is a diagonal matrix containing the singular values of $W$. To achieve column-orthogonality, we set $Q = V$ and transform $W$ as $\widehat{W} = WV$. This transformation guarantees that the resulting matrix $\widehat{W}$ satisfies the column-orthogonality:

$$(\widehat{W})^\top \widehat{W} = \Sigma^\top U^\top U \Sigma = \Sigma^2. \tag{5}$$

To ensure that the model's final output remains unchanged after this transformation, we compensate for its effects using computational invariance (Ashkboos et al., 2024a). In particular, we enforce column-wise orthogonality constraints over the matrices via SVD-based transformation. Detailed pseudoscope of Algorithm is present 1. Without loss of generality, we present pseudocode to a transformer-based model $\mathcal{M}$ equipping with $L$ layers. Each layer includes the following weight matrices: $\left\{W_k^{(\ell)}, W_q^{(\ell)}, W_v^{(\ell)}, W_o^{(\ell)}, W_{\text{gate}}^{(\ell)}, W_{\text{up}}^{(\ell)}, W_{\text{down}}^{(\ell)}\right\}$ for $\ell = 1, \ldots, L$, along with the output projection matrix $W_{\text{head}}$ of the final head layer.

**Runtime.**  This transformation is lightweight and efficient, taking less than four minutes on a single A100 GPU for models like Llama-2-3B, Llama-3-8B, and Mistral, and less than twelve minutes for Phi-4-14B.

## A.3  Results of Different Models over GSM8K

Table 10: Results of different models over GSM8K

| Sparsity | Method | Llama-2-7B | Llama-3-8B | Mistral-7B | Phi-4 (14B) |
|---|---|---|---|---|---|
| 0 | Baseline | 13.95 | 49.96 | 38.74 | 90.22 |
| 0.25 | TEAL | 14.33 | 48.98 | 37.98 | 89.84 |
|  | **WINA** | 12.51 | 49.36 | 37.04 | 90.22 |
| 0.4 | TEAL | 13.12 | 39.88 | 33.97 | 88.02 |
|  | **WINA** | 13.34 | 40.26 | 35.41 | 90.67 |
| 0.5 | TEAL | 8.72 | 27.07 | 29.49 | 86.13 |
|  | **WINA** | 11.22 | 30.40 | 29.72 | 87.57 |
| 0.65 | TEAL | 2.50 | 2.73 | 10.99 | 74.37 |
|  | **WINA** | 4.62 | 5.91 | 12.36 | 77.10 |

---

**Algorithm 1** Orthogonal Tensor Transformation

---

1: **Input:** Model $\mathcal{M}$ with matrix $W_{emb}$ of embedding layer, $L$ layers with matrices $\{W_k^{(\ell)}, W_q^{(\ell)}, W_v^{(\ell)}, W_o^{(\ell)}, W_{gate}^{(\ell)}, W_{up}^{(\ell)}, W_{down}^{(\ell)}\}_{\ell=1}^L$, and matrix $W_{head}$ of head layer.

2: **Output:** Orthogonally transformed model $\mathcal{M}'$ which is computational invariant to $\mathcal{M}$.

3: Perform SVD over $W_k^{(0)}$, $W_k^{(0)} = U\Sigma V^\top$.

4: $Q_k^{(0)} \leftarrow V$.

5: $\widehat{W}_{emb} \leftarrow W_{emb}Q_k^{(0)}$

6: **for** $\ell = 1, 2, \ldots, L$ **do**

7: $\quad \widehat{W}_k^{(\ell)} \leftarrow W_k^{(\ell)}Q_k^{(\ell)}, \quad \widehat{W}_q^{(\ell)} \leftarrow W_q^{(\ell)}Q_k^{(\ell)}, \quad \widehat{W}_v^{(\ell)} \leftarrow W_v^{(\ell)}Q_k^{(\ell)}$

8: $\quad$ Perform SVD over $W_{gate}^{(\ell)} = U\Sigma V^\top$.

9: $\quad Q_{gate}^{(\ell)} \leftarrow V$

10: $\quad \widehat{W}_o^{(\ell)} \leftarrow (Q_{gate}^{(\ell)})^\top W_o^{(\ell)}$

11: $\quad \widehat{W}_{gate}^{(\ell)} \leftarrow W_{gate}^{(\ell)}Q_{gate}^{(\ell)}, \quad \widehat{W}_{up}^{(\ell)} \leftarrow W_{up}^{(\ell)}Q_{gate}^{(\ell)}$

12: $\quad$ **if** $\ell < L$ **then**

13: $\quad\quad$ Perform SVD over $W_k^{(\ell+1)} = U\Sigma V^\top$

14: $\quad\quad Q_k^{(\ell+1)} \leftarrow V$

15: $\quad\quad \hat{W}_{down}^{(\ell)} \leftarrow (Q_k^{(\ell)})^\top W_{down}^{(\ell)}$

16: $\quad$ **end if**

17: **end for**

---

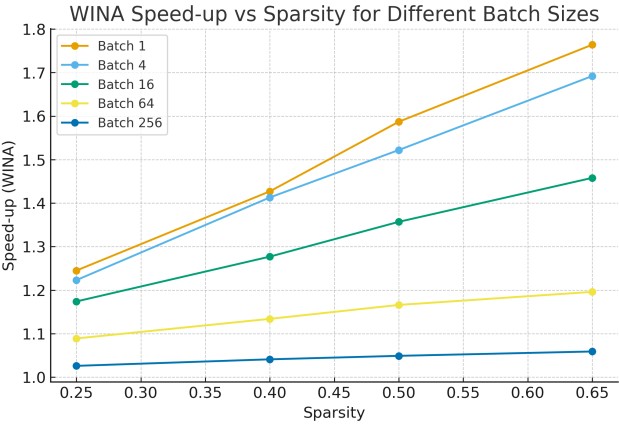

Figure 5: Speed-up ratio versus sparsity level on WINA on $5120 \times 17920$.

## A.4 SPEED-UPS ACROSS BATCH SIZES

We compute GEMV (General matrix-vector multiplication) latency and speedup via sparse activation across different batch size. The experiment is conducted on A800. We employ a matrix of shape 5120×17920, corresponding to the largest weight matrix in phi-4, the biggest LLM used in our experiments. The sequence length is set to 128. We calculate the speedup based on latency, defined as the ratio of the dense model's latency to the latency under sparse activation. The results in Figure 5 show that TEAL and WINA achieve comparable speedups at the same sparsity level, and the speedup for both decreases as the batch size increases.

## A.5 ADDITIONAL EXPERIMENTS ON LONGBENCH

Table 11: Comparison of WINA and TEAL on LongBench tasks (Llama-3-8B-8k).

| Sparsity | Method | Code Completion | Few-shot Learning | Summari-zation | Multi-Document QA | Single-Document QA | Synthetic Tasks | Overall |
|---|---|---|---|---|---|---|---|---|
| 0 | Benchmark | 23.56 | 60.36 | 16.10 | 9.87 | 13.80 | 12.76 | 23.14 |
| 0.25 | TEAL | 22.00 | 60.17 | 15.77 | 9.96 | 13.92 | 11.03 | 22.69 |
|  | WINA | 23.30 | 60.20 | 16.24 | 9.89 | 13.65 | 10.92 | **22.82** |
| 0.4 | TEAL | 21.06 | 60.46 | 15.63 | 9.55 | 13.71 | 4.74 | 21.61 |
|  | WINA | 24.13 | 60.47 | 17.71 | 9.74 | 13.02 | 7.30 | **22.57** |
| 0.5 | TEAL | 17.17 | 60.50 | 15.92 | 9.18 | 14.22 | 4.41 | 21.28 |
|  | WINA | 23.88 | 60.24 | 19.29 | 9.08 | 11.82 | 3.95 | **21.97** |
| 0.65 | TEAL | 8.29 | 51.25 | 13.86 | 7.85 | 9.78 | 3.48 | 17.05 |
|  | WINA | 19.58 | 53.61 | 14.66 | 7.07 | 7.73 | 3.07 | **18.13** |

Table 12: Comparison of WINA and TEAL on LongBench tasks (Phi-4-16k).

| Sparsity | Method | Code Completion | Few-shot Learning | Summari-zation | Multi-Document QA | Single-Document QA | Synthetic Tasks | Overall |
|---|---|---|---|---|---|---|---|---|
| 0 | Benchmark | 29.58 | 56.03 | 8.41 | 4.80 | 18.59 | 59.59 | 28.06 |
| 0.25 | TEAL | 31.16 | 54.83 | 10.11 | 5.73 | 18.32 | 55.60 | 27.86 |
|  | WINA | 30.15 | 55.53 | 9.07 | 6.48 | 19.34 | 57.46 | **28.30** |
| 0.4 | TEAL | 30.65 | 55.20 | 12.53 | 9.95 | 21.38 | 48.70 | 28.74 |
|  | WINA | 33.51 | 56.13 | 11.68 | 11.28 | 19.60 | 52.10 | **29.43** |
| 0.5 | TEAL | 29.51 | 58.44 | 15.04 | 11.69 | 22.15 | 51.03 | 30.54 |
|  | WINA | 36.53 | 59.39 | 13.39 | 15.98 | 21.19 | 48.76 | **31.39** |
| 0.65 | TEAL | 25.71 | 59.48 | 19.50 | 14.11 | 19.25 | 43.52 | 30.07 |
|  | WINA | 37.60 | 59.22 | 17.36 | 12.24 | 16.30 | 46.58 | **30.26** |

## A.6   EXTENSIVE EXPERIMENTS ON SOCIAL BIAS

Prior work (Gonçalves and Strubell, 2023) has shown that model compression can unintentionally influence social bias in LLMs. To assess whether our sparsification method exhibits similar behavior, we follow the evaluation protocol and stereotype-score metric of (Gonçalves and Strubell, 2023) and measure performance on the CrowS-Pairs benchmark across GENDER, RACE, and RELIGION categories. Table 13 reports results for our models at varying sparsity levels.

Across all models, we observe no systematic amplification of social bias as sparsity increases. Instead, bias scores typically change modestly (within 3-6 points) and often move closer to the 50% "unbiased" target defined in (Gonçalves and Strubell, 2023).

For instance, Llama-2-7B exhibits reductions in GENDER ($59.92 \rightarrow 53.82$) and RACE ($69.77 \rightarrow 66.28$) at 65% sparsity; Llama-3-8B shows mild decreases across all categories at higher sparsity levels; and Mistral-7B maintains largely stable behavior with slight improvements in RACE and RELIGION. Notably, Phi-4-14B demonstrates the strongest effect, with substantial reductions at 65% sparsity across all categories, particularly in the RELIGION category ($74.29 \rightarrow 60.00$).

Table 13: CrowS-Pairs stereotype scores for Gender, Race, and Religion for WINA over different LLMs. For each metric, we report the stereotype scores and their absolute distance from the optimal unbiased value of 50% in parenthesis (smaller values indicate less bias from neutrality,). Scores further from 50% indicate more bias. Green indicates improved bias scores from baseline, red indicates worse, and grey indicates no change from baseline.

| Model | Sparsity | GENDER | RACE | RELIGION |
|---|---|---|---|---|
| Llama-2-7B | Baseline | 59.92 (9.92) | 69.77 (19.77) | 74.29 (24.29) |
|  | 25% | 59.92 (9.92) | 67.83 (17.83) | 77.14 (27.14) |
|  | 40% | 58.40 (8.40) | 71.51 (21.51) | 78.10 (28.10) |
|  | 50% | 59.54 (9.54) | 68.02 (18.02) | 74.29 (24.29) |
|  | 65% | 53.82 (3.82) | 66.28 (16.28) | 76.19 (26.19) |
| Llama-3-8B | Baseline | 60.31 (10.31) | 66.28 (16.28) | 74.29 (24.29) |
|  | 25% | 59.54 (9.54) | 65.70 (15.70) | 76.19 (26.19) |
|  | 40% | 61.45 (11.45) | 66.09 (16.09) | 76.19 (26.19) |
|  | 50% | 57.63 (7.63) | 65.50 (15.50) | 71.43 (21.43) |
|  | 65% | 60.69 (10.69) | 64.34 (14.34) | 68.57 (18.57) |
| Mistral-7B | Baseline | 62.98 (12.98) | 67.25 (17.25) | 69.52 (19.52) |
|  | 25% | 62.60 (12.60) | 68.22 (18.22) | 66.67 (16.67) |
|  | 40% | 62.21 (12.21) | 64.53 (14.53) | 68.57 (18.57) |
|  | 50% | 62.60 (12.60) | 67.05 (17.05) | 69.52 (19.52) |
|  | 65% | 61.83 (11.83) | 64.15 (14.15) | 64.76 (14.76) |
| Phi-4-14B | Baseline | 65.65 (15.65) | 63.95 (13.95) | 74.29 (24.29) |
|  | 25% | 63.36 (13.36) | 63.57 (13.57) | 71.43 (21.43) |
|  | 40% | 61.83 (11.83) | 60.66 (10.66) | 67.62 (17.62) |
|  | 50% | 59.54 (9.54) | 61.43 (11.43) | 65.71 (15.71) |
|  | 65% | 59.54 (9.54) | 59.88 (9.88) | 60.00 (10.00) |

Taken together, our results suggest that our sparsification method does not exacerbate social bias and in some cases modestly reduces it, consistent with the behavior of certain compression settings reported in (Gonçalves and Strubell, 2023): all in all, we see reduced bias in 12/16 experiments for GENDER, 14/16 for RACE, and 9/16 for RELIGION. We emphasize, however, that these benchmarks capture only a limited set of bias dimensions. We view these findings as indicative rather than comprehensive, and further research on fairness-aware compression remains an important direction.

### A.7  PROOF OF LEMMA 3.1

**Proof.**    Let $\mathcal{I}^{=0}(\boldsymbol{x}) := \{i|\boldsymbol{x}_i = 0\}$ be the set of indices of zero elements at $\boldsymbol{x}$. The output deviation between the original network output and the gated output via a general-format sparsification is:

$$
\begin{aligned}
\|W(\boldsymbol{x}_{\mathcal{I}^{=0}} - \boldsymbol{x})\|^2 &= \left\| \sum_{i \in \mathcal{I}^{=0}} \boldsymbol{x}_i W_{:,i} \right\|_2^2 \\
&= \left( \sum_{i \in \mathcal{I}^{=0}} x_i W_{:,i} \right)^{\top} \left( \sum_{i \in \mathcal{I}^{=0}} x_i W_{:,i} \right) \\
&= \sum_{j \in \mathcal{I}^{=0}} \sum_{i \in \mathcal{I}^{=0}} \boldsymbol{x}_j \boldsymbol{x}_i W_{:,j}^{\top} W_{:,i} \\
&= \sum_{i \in \mathcal{I}^{=0}} \boldsymbol{x}_i^2 \|W_{:,i}\|_2^2 + \sum_{i \ne j \in \mathcal{I}^{=0}} \boldsymbol{x}_j \boldsymbol{x}_i W_{:,j}^{\top} W_{:,i}
\end{aligned}
$$

The expected output deviation for WINA is:

$$
\begin{aligned}
e_{\text{WINA}} &= \left\| W \boldsymbol{x}_{\mathcal{I}_{\text{WINA}}^{=0}} - W\boldsymbol{x} \right\|^2 \\
&= \sum_{i \in \mathcal{I}_{\text{WINA}}^{=0}} \boldsymbol{x}_i^2 \|W_{:,i}\|_2^2 + \sum_{i \ne j \in \mathcal{I}_{\text{WINA}}^{=0}} \boldsymbol{x}_j \boldsymbol{x}_i W_{:,j}^{\top} W_{:,i}.
\end{aligned}
$$

Since $W$ is assumed to be column orthogonal, the cross-term expectations vanish, and the expected output error is determined solely by the main term:

$$
e_{\text{WINA}} = \sum_{i \in \mathcal{I}_{\text{WINA}}^{=0}} \boldsymbol{x}_i^2 \|W_{:,i}\|_2^2.
$$

Because WINA sparsification sets the $k$ smallest $|\boldsymbol{x}_i \boldsymbol{c}_i|$ terms to zero, we have the mask of WINA reaches out the lower bound of approximation error for a single layer network, i.e.,

$$
\boldsymbol{g}_{\text{WINA}}(\boldsymbol{x}) = \underset{\boldsymbol{g} \in \{0,1\}^n}{\arg\min} \quad \|W(\boldsymbol{x} \odot \boldsymbol{g} - \boldsymbol{x})\|^2 . \tag{6}
$$

Thus, the above indicates that WINA sparsification achieves the tight lower bound of the approximation error, including those of TEAL and CATS.

### A.8  PROOF OF THEOREM 3.2

**Proof.**    We consider an $L$-layer *linear* network (no activation):

$$
\boldsymbol{y}^{(1)} = W^{(1)}\boldsymbol{x}, \qquad \boldsymbol{y}^{(\ell+1)} = W^{(\ell+1)}\boldsymbol{y}^{(\ell)}, \ \ell = 1, \ldots, L-1,
$$

so

$$
\mathcal{M}(\boldsymbol{x}) = W^{(L)} \cdots W^{(1)} \boldsymbol{x}.
$$

Let $\boldsymbol{g}^{(\ell)} \in \{0,1\}^{d_\ell}$ be a gate and define $M^{(\ell)} = \operatorname{diag}(\boldsymbol{g}^{(\ell)})$. The gated network is defined recursively by

$$
\boldsymbol{y}_{\boldsymbol{g}}^{(\ell)} = W^{(\ell)}\big(M^{(\ell)}\boldsymbol{y}_{\boldsymbol{g}}^{(\ell-1)}\big), \qquad \boldsymbol{y}_{\boldsymbol{g}}^{(0)} = \boldsymbol{x}.
$$

Denote $\mathcal{G} = \{\boldsymbol{g}^{(1)}, \ldots, \boldsymbol{g}^{(L)}\}$.

**Key assumption.**    For each layer $\ell \ge 2$, assume *column-orthogonality*:

$$
W^{(\ell)\top} W^{(\ell)} = D^{(\ell)} \text{ is diagonal.}
$$

(Equivalently, columns of $W^{(\ell)}$ are mutually orthogonal; no normalization is required.)

**Step 1: Base case** $N = 2$ **and exact decomposition.** For two layers,

$$\boldsymbol{y}^{(2)} = W^{(2)}W^{(1)}\boldsymbol{x}, \qquad \boldsymbol{y}_g^{(2)} = W^{(2)}M^{(2)}W^{(1)}M^{(1)}\boldsymbol{x}.$$

Let the deviation be $\boldsymbol{e}_g^{(2)} := \boldsymbol{y}_g^{(2)} - \boldsymbol{y}^{(2)}$. Add and subtract $W^{(2)}W^{(1)}M^{(1)}\boldsymbol{x}$:

$$\boldsymbol{e}_g^{(2)} = \underbrace{W^{(2)}(M^{(2)} - I)W^{(1)}M^{(1)}\boldsymbol{x}}_{:=\boldsymbol{v}} + \underbrace{W^{(2)}W^{(1)}(M^{(1)} - I)\boldsymbol{x}}_{:=\boldsymbol{u}}.$$

Therefore,

$$\|\boldsymbol{e}_g^{(2)}\|_2^2 = \|\boldsymbol{u}\|_2^2 + \|\boldsymbol{v}\|_2^2 + 2\,\boldsymbol{u}^\top\boldsymbol{v}.$$

This identity is exact.

**Separable expressions for** $\|\boldsymbol{u}\|_2^2$ **and** $\|\boldsymbol{v}\|_2^2$. Let $D^{(2)} := W^{(2)\top}W^{(2)}$ which is diagonal by assumption. Write $\boldsymbol{u} = W^{(2)}\boldsymbol{a}$ and $\boldsymbol{v} = W^{(2)}\boldsymbol{b}$ where

$$\boldsymbol{a} := W^{(1)}(M^{(1)} - I)\boldsymbol{x}, \qquad \boldsymbol{b} := (M^{(2)} - I)W^{(1)}M^{(1)}\boldsymbol{x}.$$

Then

$$\|\boldsymbol{u}\|_2^2 = \boldsymbol{a}^\top D^{(2)}\boldsymbol{a}, \qquad \|\boldsymbol{v}\|_2^2 = \boldsymbol{b}^\top D^{(2)}\boldsymbol{b}.$$

Because $D^{(2)}$ is diagonal and $(M^{(2)} - I)$ keeps exactly the coordinates with $g_j^{(2)} = 0$,

$$\|\boldsymbol{v}\|_2^2 = \sum_{j:g_j^{(2)}=0} \|W_{:,j}^{(2)}\|_2^2 \cdot \big((W^{(1)}M^{(1)}\boldsymbol{x})_j\big)^2.$$

Moreover, letting $A := W^{(2)}W^{(1)}$ and noting $\boldsymbol{u} = A(M^{(1)} - I)\boldsymbol{x}$, we have

$$\|\boldsymbol{u}\|_2^2 = \sum_{i:g_i^{(1)}=0} \|A_{:,i}\|_2^2 \cdot x_i^2.$$

The cross-term equals

$$\boldsymbol{u}^\top\boldsymbol{v} = \boldsymbol{a}^\top D^{(2)}\boldsymbol{b} = \sum_j D_{jj}^{(2)}\,\boldsymbol{a}_j\,\boldsymbol{b}_j,$$

which in general *depends on the gates* and cannot be assumed to vanish without additional structure.

**A provable upper bound.** For any $\alpha > 0$, by Young's inequality,

$$2\,\boldsymbol{u}^\top\boldsymbol{v} \le \alpha\|\boldsymbol{u}\|_2^2 + \frac{1}{\alpha}\|\boldsymbol{v}\|_2^2,$$

hence

$$\|\boldsymbol{e}_g^{(2)}\|_2^2 \le (1+\alpha)\|\boldsymbol{u}\|_2^2 + \left(1 + \frac{1}{\alpha}\right)\|\boldsymbol{v}\|_2^2.$$

Define the upper bound objective

$$\mathcal{U}_\alpha(\boldsymbol{x}; \mathcal{G}) := (1+\alpha)\|\boldsymbol{u}\|_2^2 + \left(1 + \frac{1}{\alpha}\right)\|\boldsymbol{v}\|_2^2.$$

Since $\|\boldsymbol{u}\|_2^2$ and $\|\boldsymbol{v}\|_2^2$ have separable (top-$k$) forms above under column-orthogonality, WINA's selection rule minimizes $\mathcal{U}_\alpha(\boldsymbol{x}; \mathcal{G})$.

**Step 2: Inductive extension to** $N > 2$. For $N + 1$ layers, let the deviation be $\boldsymbol{e}_g^{(N+1)} := \boldsymbol{y}_g^{(N+1)} - \boldsymbol{y}^{(N+1)}$. Add and subtract $W^{(N+1)}\boldsymbol{y}_g^{(N)}$:

$$\boldsymbol{e}_g^{(N+1)} = \underbrace{W^{(N+1)}(M^{(N+1)} - I)\boldsymbol{y}_g^{(N)}}_{:=\boldsymbol{v}} + \underbrace{W^{(N+1)}(\boldsymbol{y}_g^{(N)} - \boldsymbol{y}^{(N)})}_{:=\boldsymbol{u}}.$$

Thus,

$$\|\boldsymbol{e}_g^{(N+1)}\|_2^2 = \|\boldsymbol{u}\|_2^2 + \|\boldsymbol{v}\|_2^2 + 2\,\boldsymbol{u}^\top\boldsymbol{v},$$

and applying Young's inequality yields

$$\|e_g^{(N+1)}\|_2^2 \le (1+\alpha)\|u\|_2^2 + \left(1 + \frac{1}{\alpha}\right)\|v\|_2^2 =: \mathcal{U}_\alpha(x; \mathcal{G}).$$

Under the column-orthogonality assumption for $W^{(N+1)}$, $\|v\|_2^2$ is separable:

$$\|v\|_2^2 = \sum_{j: g_j^{(N+1)}=0} \|W_{:,j}^{(N+1)}\|_2^2 \cdot \left(y_{g,j}^{(N)}\right)^2,$$

so minimizing it reduces to keeping the $k$ largest weighted coordinates. Moreover, $\|u\|_2^2 = \|W^{(N+1)}(y_g^{(N)} - y^{(N)})\|_2^2$ is minimized by recursively applying the same rule to the first $N$ layers. Therefore WINA minimizes the upper bound $\mathcal{U}_\alpha(x; \mathcal{G})$ for all $L$.

The above establishes that, under column-orthogonality, WINA optimizes a provable upper bound $\mathcal{U}_\alpha(x; \mathcal{G})$ on the true deviation $\|\mathcal{M}(x) - \mathcal{M}(x \mid \mathcal{G})\|_2^2$. In general, without additional assumptions forcing $u^\top v = 0$ (or gate-independent), one cannot claim exact optimality for the true deviation. $\square$

## A.9 LIMITATIONS

To maximize performance gains and ensure realistic acceleration, WINA requires a few additional operations. First, WINA needs the column-wise norms of the weight matrix $W$, which can be pre-computed once during model loading and reused throughout inference. Second, WINA performs an element-wise product between the input and the pre-computed norms; this operation is extremely lightweight and adds negligible overhead.

The performance benefits of WINA are maximized when the weight matrices in LLMs satisfy the column-wise orthogonality condition. To enforce this property, we adopt the efficient one-off orthogonalization transformation proposed in (Ashkboos et al., 2024a). This transformation is applied a single time prior to inference and produces a numerically equivalent LLMs for further usages.

## A.10 ADDITIONAL RESULTS ON HARDWARE ACCELERATION

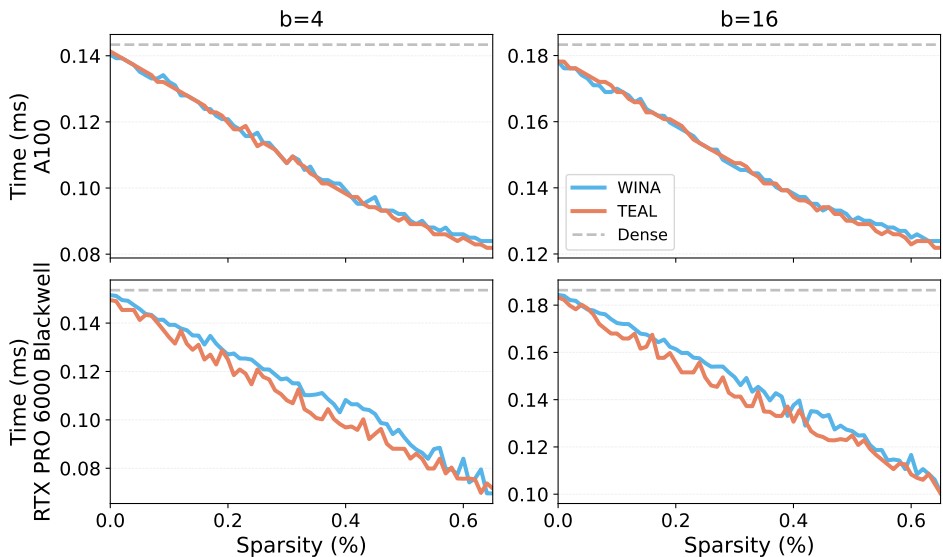

Figure 6: 5120x17920 for batch sizes 1 and 4 on A100 and RTX PRO 6000 Blackwell.

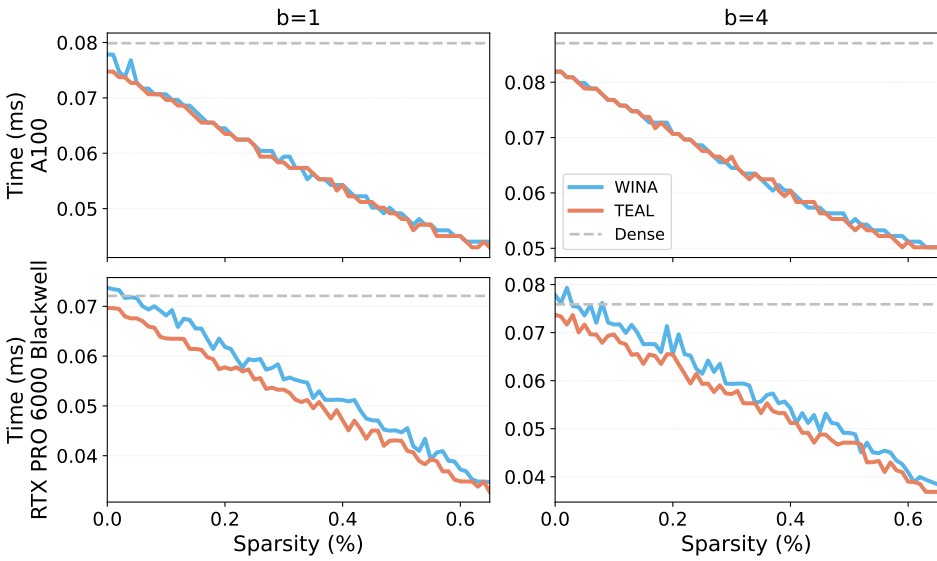

Figure 7: 4096x11008 GEMV for batch sizes 1 and 4 on A100 and RTX PRO 6000 Blackwell.

## A.11 NOTE ON LAYER-WISE SPARSITY ALLOCATION

Additionally, in this work, we do not focus specifically on per-layer sparsity allocation since WINA is generalized well and applicable over varying sparsity allocations. As future work, we believe dedicated sparsity-allocation strategies and a focus on layer-wise sparsity using different sparsity assignment protocols are promising directions to further improve WINA's performance.

