# OpenReview forum: "WINA: Weight Informed Neuron Activation for Accelerating Large Language Model Inference"
_ICLR.cc/2026/Conference — ICLR 2026 Poster_

### Official Review · Reviewer_iVmy · 2025-10-31

**Soundness:** 3
**Presentation:** 3
**Contribution:** 3
**Rating:** 6
**Confidence:** 4

**Summary:**

This paper proposed a framework WINA for training-free sparse activation that incorporates both hidden state magnitudes and weight matrix structure, which combines the magnitude of activations with the column-wise norm of the weight matrices to preserve the top-k activations. The authors claimed that WINA can achieve a lower approximation error bound under several assumptions and is model-agnostic. Methods are tested on Llama-2-7B, Llama-3-8B, Mistral-7B, and Phi-4-14B models across several benchmark datasets, which demonstrate that WINA can achieve superior performance under various sparsity ratios.

**Strengths:**

1. The combinatorial gating strategy is reasonable, which produces a tighter approximation error bound.
2. WINA as a training-free method is friendly for deployment.
3. The paper is well written and easy to follow.

**Weaknesses:**

1. There are no end-to-end latency performance comparisons between WINA and previous methods like TEAL, CATS, and R-Sparse.
2. For the math reasoning task like GSM8K, the aggressive sparsity can induce significant damage to the model performance, dropping accuracy from 50 to 7, although WINA reported superior performance to the baseline methods.
3. When the batch size is larger, will the sparsity be affected and further the speedup gain be degraded?

**Questions:**

See weaknesses.

---

> ### Author Response · Authors · 2025-11-20
> **Response to reviewer iVmy**
>
> We appreciate the Reviewer iVmy for the constructive review and comments. We provide our detailed answers to questions/concerns below.
>
> **Q1.  There are no end-to-end latency performance comparisons between WINA and previous methods like TEAL, CATS, and R-Sparse.**
> > A1. Thanks for the important question. We have developed a dedicated WINA Triton kernel and benchmarked it extensively across multiple GPU devices (A100, A800, and RTX-6000 Blackwell), a wide range of batch sizes (1–256) (more discussions in General Response Q3/A3 at the top), and diverse matrix dimensions. The experiments show that WINA delivers nearly identical acceleration and wall-clock speed-up to TEAL relative to the dense baseline, confirming that WINA provides practical and realistic efficiency.
> >
> > We include the detailed latency curves in General Response Q1/A1, Section 4.4, and Appendix A.4 and A.11 of the revised manuscript. Importantly, because WINA achieves significantly better model performance at the same sparsity, it also provides greater effective acceleration at a fixed accuracy level compared to prior state-of-the-art methods such as TEAL.
>
>
> **Q2. For the math reasoning task like GSM8K, the aggressive sparsity can induce significant damage to the model performance, dropping accuracy from 50 to 7, although WINA reported superior performance to the baseline methods.**
> > A2. Thank you for the thoughtful observation. We agree that GSM8K is sometimes more sensitive to high sparsity. We observe that the degree of degradation depends on the underlying base model. Different models exhibit varying robustness under sparse activation.
> >In our experiments, Llama-2-7B and Llama-3-8B show noticeable degradation at high sparsity, whereas Phi-4-14B maintains substantially stronger performance, indicating base model differences in resilience to sparsity on GSM8K.
> >
> > Importantly, across almost all architectures and sparsity levels, WINA consistently outperforms TEAL, achieving better performance preservation. The table below summarizes the GSM8K results across models and sparsity levels
> >
> >| GSM8K | Sparsity | Llama-2-7B | Llama-3-8B | Mistral-7B | Phi-4-14B |
> >|-----------|----------|------------|------------|------------|-----------|
> >| Baseline | 0 | 0.1395 | 0.4996 | 0.3874 | 90.22 |
> >| **WINA** | 0.25 | 0.1251 | **0.4936** | 0.3704 | **90.22** |
> >| | 0.4 | **0.1334** | **0.4026** | **0.3541** | **90.67** |
> >| | 0.5 | **0.1122** | **0.3040** | **0.2972** | **87.57** |
> >| | 0.65 | **0.0462** | **0.0591** | **0.1236** | **77.10** |
> >| **TEAL** | 0.25 | **0.1433** | 0.4898 | **0.3798** | 89.84 |
> >| | 0.4 | 0.1312 | 0.3988 | 0.3397 | 88.02 |
> >| | 0.5 | 0.0872 | 0.2707 | 0.2949 | 86.13 |
> >| | 0.65 | 0.0250 | 0.0273 | 0.1099 | 74.37 |
>
>
> **Q3. When the batch size is larger, will the sparsity be affected and further the speedup gain be degraded?**
> > A3. Thank you for the insightful question. Based on our experiments using the custom WINA Triton kernels across multiple GPU types, we observe that the speedup remains stable as batch size increases, as long as the workload does not saturate GPU memory.
> > In other words, sparsity-induced acceleration is not degraded by larger batch sizes until the GPU reaches its memory limit. Once the memory becomes fully occupied, the speedup saturates due to hardware constraints. We note that TEAL reported similar phenomenon.
> > More detailed results are provided in Figure 5 of A.4 of our revised manuscript.
> >
> > **Latency and speedup ratio across different batch size for WINA)**
> >| Sparsity | Batch Size 1 | Batch Size 4 | Batch Size 16 | Batch Size 64 | Batch Size 256 |
> >|--|--|--|--|--|--|
> >| 0 | 1x | 1x | 1x | 1x | 1x |
> >| 0.25 | 1.245x | 1.223x | 1.174x | 1.089x | 1.026x |
> >| 0.4 | 1.427x | 1.413x | 1.277x | 1.134x | 1.039x |
> >| 0.5 | 1.587x | 1.522x | 1.357x | 1.166x | 1.049x |
> >| 0.65  | 1.764x | 1.692x | 1.458x | 1.196x | 1.059x |

---

### Official Review · Reviewer_6SUJ · 2025-11-02

**Soundness:** 3
**Presentation:** 3
**Contribution:** 2
**Rating:** 4
**Confidence:** 3

**Summary:**

This paper presents WINA (Weight-Informed Neuron Activation), a training-free sparse activation method that accelerates LLM inference by selecting neurons based on both hidden state magnitudes and weight matrix norms. This weight-informed approach achieves tighter theoretical error bounds and better accuracy than prior methods like TEAL and CATS, maintaining strong performance even at high sparsity levels across various LLMs and tasks, and showing compatibility with quantized models.

**Strengths:**

1. The paper is well-organized and easy to follow.

2. The figures are clearly and beautifully presented.

3. The experiments conducted on extensive datasets provide strong validation and demonstrate the integrity of the proposed method.

**Weaknesses:**

1. My main concern is related to the performance measurement. The authors claim that WINA is more efficient than previous methods, "potentially translating to faster inference speeds and lower computational costs." Could the authors provide empirical evidence, such as wall-clock time or GPU memory usage, to support this claim?

2. The improvement in synthetic results shown in Table 2 is substantial, but the gains in real-world LLM experiments are relatively modest. Could the authors clarify the reason for this large discrepancy between synthetic and real-world results?

3. How can the assumption of "column-wise orthogonality" in the theorems be verified? Is there any experimental evidence to support this assumption?

**Questions:**

See weaknesses above.

---

> ### Author Response · Authors · 2025-11-20
> **Response to reviewer 6SUJ**
>
> We thank Reviewer 6SUJ for the constructive review and comments. We provide our detailed answers to each question/point in turn below.
>
> **Q1. My main concern is related to the performance measurement. The authors claim that WINA is more efficient than previous methods, "potentially translating to faster inference speeds and lower computational costs." Could the authors provide empirical evidence, such as wall-clock time or GPU memory usage, to support this claim?**
> > A1. Thank you for highlighting this important point.  To clarify our claim: under the same sparsity level, WINA achieves (significantly) higher performance than prior training-free sparse activation methods. This is supported by both our theoretical analysis and extensive empirical results.
> >
> > To directly address your concern about realistic efficiency, we additionally implemented a custom Triton-based WINA kernel and measured wall-time throughput across varying sparsities and multiple GPU devices. These results are now included in General Response Q1/A1, Section 4.4, and Appendix A.4 and A.11 in the revised version of our manuscript. They show that WINA would achieve almost identical realistic wall-time compared to TEAL.
> > This directly implies that for the same performance target, WINA can achieve
> > -   higher effective inference speed,
> > -   lower computational cost,
> > -   greater FLOPs savings
> >
> > compared with previous methods.
>
> **Q2. The improvement in synthetic results shown in Table 2 is substantial, but the gains in real-world LLM experiments are relatively modest. Could the authors clarify the reason for this large discrepancy between synthetic and real-world results?**
> > A2. This is a great question. The reasons are two-fold.
> >
> > First, the evaluation metrics differ between the two settings. Table 2 measures $\ell_2$ output deviation, which WINA is specifically designed to minimize. By contrast, the downstream evaluations in lm-evaluation-harness measure task-level performance, which depends on additional factors beyond approximation error (e.g., attention dynamics, decoding behavior). This metric mismatch might lead to different observable improvements at the task level.
> >
> > Secondly, In Table 2 (Sec. 3.4), the synthetic experiments are constructed so that all assumptions behind our theoretical analysis are fully satisfied. Under these controlled conditions, WINA aligns exactly with the tighter approximation rule, leading to the substantial improvements observed. In contrast, real-world LLMs might violate some ideal assumptions, such as residual paths and self-attention create complex dependencies (e.g., specialized architecture), etc. As a result, while WINA remains principled and effective, its theoretical optimality might not be fully achieved in practice, which reduces the magnitude of gains to an extent but such gains remain significant.
>
>
> **Q3. How can the assumption of "column-wise orthogonality" in the theorems be verified? Is there any experimental evidence to support this assumption?**
> > A3. Thanks for the great question. We provided a general response regarding that. To enforce column-wise orthogonality, there are two families of approaches, one is adding orthogonality regularizer into loss function [1-2], the other one is via matrix transformation [3-7].  In our work, we adopt the second approach because it can be applied once offline to obtain a numerically equivalent LLM, without additional training or fine-tuning. Since these methods are already well established in prior literature, we have updated the manuscript to include and clarify these approaches.
> >
> > [1] All You Need is Beyond a Good Init: Exploring Better Solution for Training Extremely Deep Convolutional Neural Networks with Orthonormality and Modulation.
> >
> > [2] Controllable Orthogonalization in Training DNNs.
> >
> > [3] SliceGPT: Compress Large Language Models by Deleting Rows and Columns.
> >
> > [4] QuaRot: Outlier-Free 4-Bit Inference in Rotated LLMs.
> >
> > [5] QuiP: 2-Bit Quantization of Large Language Models with Guarantees.
> >
> > [6] SpinQuant: LLM Quantization with learned rotations.
> <
> > [7] OstQuant: Refining Large Language Model Quantization with Orthogonal and Scaling Transformations for Better Distribution Fitting.
>
> We hope that our responses and the newly added Triton-kernel experiments adequately address the raised concerns. We would be grateful if you could kindly reconsider the overall assessment in light of these strengthened results and increase the rating if appropriate.

---

> > ### Comment · Reviewer_6SUJ · 2025-11-25
> >
> > Thanks to the author for the response, which addressed my concerns. I have therefore decided to raise my score.

---

> > > ### Author Response · Authors · 2025-11-26
> > >
> > > Thank you again for your questions, suggestions, and overall review. We very much appreciate the time and consideration given to our paper and your help in improving our work.

---

### Official Review · Reviewer_1iPj · 2025-11-03

**Soundness:** 3
**Presentation:** 3
**Contribution:** 3
**Rating:** 6
**Confidence:** 3

**Summary:**

The paper aims to improve inference efficiency of LLMs and proposes skipping unnecessary neuron computations in feed-forward network (FFN) layers. The proposed method does this with a gating function that uses the magnitude of a neuron's output weight to assess its importance. Less important neurons (smaller output weight magnitue and smaller intermediate activation) are skipped. The paper has empirical results on Llama 3-8B and Qwen 2-7B, demonstrating significant speedups with minimal impact on model accuracy.

**Strengths:**

- The proposed method, WINA, is easy to apply to existing pre-trained LLMs since it is a post-training method that does not require any fine-tuning or retraining.

- To the best of my knowledge, assigning importance scores based on output weight's magnitude is a novel idea.

- Empirical results are strong across various benchmarks and model sizes.

**Weaknesses:**

- The linked source code is not available.

- The success of WINA relies on the threshold used to determine which neurons to skip. Tuning this threshold would be costly.

**Questions:**

Do the authors have any suggestions on how to tune the threshold efficiently?

---

> ### Author Response · Authors · 2025-11-20
> **Response to reviewer 1iPj**
>
> We appreciate Reviewer 1iPj for the constructive review and comments. We provide our detailed answers in response to the questions and suggestions below.
>
> **Q1. The linked source code is not available.**
> > A1. Thanks for the question. We double checked the link of source code provided in abstract and verified that it is functional and accessible. For convenience, we also attach the source code in the supplementary file in the revision.
>
> **Q2. The success of WINA relies on the threshold used to determine which neurons to skip. Tuning this threshold would be costly.**
> > A2. We would like to gently clarify that WINA does not rely on manually tuned thresholds or layer-specific sparsity heuristics.  The key improvement comes from the weight-informed activation mechanism, which replaces the purely activation-magnitude criterion used in prior work to reduce approximation error.
> In all our experiments, we adopt exactly the same sparsity allocation protocol as TEAL, yet WINA consistently achieves substantially better performance, especially at high sparsity. It demonstrates that the gain comes from the proposed gating mechanism rather than from threshold tuning.
>
> **Q3. Do the authors have any suggestions on how to tune the threshold efficiently?**
> > A3. We strongly agree with the reviewer that dedicated sparsity-allocation strategies could further improve performance, though is not the focus on current work. In practice, we recommend the following workflow:
> > - **Determine the target sparsity or desired speedup**, using the sparsity–wall-time curves we provide for WINA in the revision.
> > - **Allocate layer-wise sparsity** using different sparsity assignment protocols, such as the one TEAL provided.
> >

---

> > ### Comment · Reviewer_1iPj · 2025-11-25
> >
> > Thank you for the response.

---

> ### Author Response · Authors · 2025-11-26
>
> Thank you again for your constructive comments. We greatly appreciate your feedback which has strengthened our work. If any additional questions arise, please feel free to let us know and we would be happy to provide further details during the remainder of the rebuttal period.

---

### Official Review · Reviewer_DJxS · 2025-11-03

**Soundness:** 2
**Presentation:** 3
**Contribution:** 2
**Rating:** 6
**Confidence:** 3

**Summary:**

The paper proposes a new sparse activation method, WINA, to improve LLM inference efficiency. WINA or Weight Informed Neuron Activation uses both the hidden state magnitude and weight matrix structure while sparsifying the activation; previous work only relies on the hidden magnitude.
WINA uses the product of l2 norm of the column vector of the weight matrix and the hidden state magnitude to select the top-K neuron with theoretical justification. Extensive experiments are provided with popular benchmark datasets.

**Strengths:**

1. The paper is well-written and easy to follow, and the idea is intuitive.

2. Theoretical justification showing that using both the L2 norm of the column vector and the hidden state magnitude yields an optimal solution and reduces error (section 3).

3. Results provided are quite extensive; the method is evaluated on multiple datasets and different downstream tasks.

4. Results in tables 3 and 4 show consistent improvements, especially at higher sparsities, compared to other baselines, showing the efficacy of the method.

5. Additional results showing on quantization are provided, showing WINA is compatible with quantization.

**Weaknesses:**

1. The technical novelty of the method is limited (however, results show it improves over baselines).

2. It is not clear why the orthogonality of the weight matrix is enforced (in sec 3.4)? Does this orthogonality hold in a general setting as well?

3. Recent works have shown that LLM compression has an unintended impact on the model bias. It would be helpful to also evaluate the impact of the proposed method on model bias.


[1]. Strubell et al., Understanding the Effect of Model Compression on Social Bias in Large Language Models

**Questions:**

1. How is LayerNorm monotonically increasing? (Line 238)

---

> ### Author Response · Authors · 2025-11-20
> **Response to reviewer DJxS**
>
> We appreciate Reviewer DJxS for their constructive review and comments. We provide our detailed answers in line with each  of their questions and suggestions below.
>
> **Q1. The technical novelty of the method is limited while the improvement is significant.**
> > A1. Thank you for acknowledging our strong empirical results. WINA is indeed a simple and effective gating mechanism, but we would like to gently point out that WINA's simplicity does not diminish its technical contribution. Because the WINA activation rule is principle-driven, derived from minimizing the layer-wise approximation error under sparsity constraints, formulated as a closed-solution, and is supported by theoretical guarantee as well as extensive numerical validation. These aspects, together, are their own technical novelties compared to previous approaches.
>
> **Q2. It is not clear why the orthogonality of the weight matrix is enforced (in sec 3.4)? Does this orthogonality hold in a general setting as well?**
> > A2. Thank you for the insightful question.
> > In Section 3.4, we create a setting where the assumptions of theorem holds  allowing us to derive and validate the tighter approximation error for WINA. To make that happen, we randomly initialize weight matrix $W$ which is then fed into the orthogonality transformation to produce orthogonal $\hat{W}$ in Eq. (5) of Appendix A.2. In particular, we proceed with a singular value decomposition $W=U\Sigma V^\top$, where $U$ and $V$ are orthogonal, then $\hat{W}=W V=U\Sigma V^\top V=U\Sigma$. Consequently, $\hat{W}$ is orthogonal since $\hat{W}^\top \hat{W}=\Sigma^\top U^\top U\Sigma=I$.
> >
> >
> > In general setting, weight matrices in standard neural networks are not orthogonal. However, prior works have proposed two families of approaches, one of which can transform a given network into a numerically equivalent counterpart inspired by the above transformation.  Following this line of transformation, we apply an orthogonality transformation proposed by [1] to maximize the performance gains of WINA. For more information, we refer to point Q2/A2 of our General Response at the top.
> >
> > [1] SliceGPT: Compress Large Language Models by Deleting Rows and Columns.

---

> ### Author Response · Authors · 2025-11-20
> **Continued response to reviewer DJxS**
>
> **Q3. Recent works have shown that LLM compression has an unintended impact on the model bias. It would be helpful to also evaluate the impact of the proposed method on model bias.**
> > A3. Thank you for the excellent suggestion. Following the reviewer’s guidance, we evaluated the impact of WINA on social bias. We applied WINA to Llama-2-7B, Llama-3-8B, Mistral-7B, and Phi-4-14B, and measured bias across Gender, Race, and Religion dimensions using **BiasBench** [2] under varying sparsity levels.
> > To explicitly quantify the effect on bias, we followed the experimental setup and stereotype score computation of [2] on the CrowS-Pairs benchmark (GENDER, RACE, RELIGION; lower scores closer to 50% indicate less biased behavior). We have updated our paper to include these results with a discussion in Appendix A.4 but also pasted our experimental results below for convenience.**
> >
> > |Model | Sparsity | Gender | Race | Religion |
> > |--|--|--|--|--|
> > | **Llama 2 7B** | Baseline | 59.92 | 69.77 | 74.29 |
> > | | 25% | 59.92 | 67.83 | 77.14 |
> > | | 40% | 58.40 | 71.51 | 78.10 |
> > | | 50% | 59.54 | 68.02 | 74.29 |
> > | | 65% | 53.82 | 66.28 | 76.19 |
> > | **Llama 3 8B** | Baseline | 60.31 | 66.28 | 74.29 |
> > | | 25% | 59.54 | 65.70 | 76.19 |
> > | | 40% | 61.45 | 66.09 | 76.19 |
> > | | 50% | 57.63 | 65.50 | 71.43 |
> > | | 65% | 60.69 | 64.34 | 68.57 |
> > | **Mistral 7B** | Baseline | 62.98 | 67.25 | 69.52 |
> > | | 25% | 62.60 | 68.22 | 66.67 |
> > | | 40% | 62.21 | 64.53 | 68.57 |
> > | | 50% | 62.60 | 67.05 | 69.52 |
> > | | 65% | 61.83 | 64.15 | 64.76 |
> > | **Phi 4 14B** | Baseline | 65.65 | 63.95 | 74.29 |
> > | | 25% | 63.36 | 63.57 | 71.43 |
> > | | 40% | 61.83 | 60.66 | 67.62 |
> > | | 50% | 59.54 | 61.43 | 65.71 |
> > | | 65% | 59.54 | 59.88 | 60.00 |
> >
> > Overall, we do not observe systematic bias amplification as sparsity increases. Under our methodology, changes are modest and can often reduce stereotype scores relative to the dense/original model baseline:
> > * Llama 2 7B. Increasing sparsity from dense to 65% reduces the GENDER score from 59.92 to 53.82 (-6.1 points, closer to the 50% “unbiased” score) and RACE from 69.77 to 66.28 (-3.5), while RELIGION is roughly stable to slightly higher (74.29 to 76.19).
> > * Llama 3 8B. Sparsity has a small regularizing effect across all categories. At 50-65% sparsity, the GENDER score drops from 60.31 to 57.63, RACE from 66.28 to 64.34, and RELIGION from 74.29 to 68.57 (-5.7 points).
> > * Mistral 7B. We again see small changes, with 65% sparsity reducing RACE (67.25 to 64.15) and RELIGION (69.52 to 64.76), while GENDER remains relatively stable (62.98 to 61.83).
> > * Phi 4 14B. Here the effect is strongest: going from dense to 65% sparsity consistently reduces bias across categories, with GENDER 65.65 to 59.54 (-6.1), RACE 63.95 to 59.88 (-4.1), and RELIGION 74.29 to 60.00 (-14.3).
> >
> > We have added these CrowS-Pairs results to the paper (Appendix A.5) and explicitly discuss that, under the stereotype score metric of [2], our sparsification method does not exacerbate social bias and can modestly reduce it. At the same time, we note that these benchmarks cover only three bias dimensions and should be viewed as indicative rather than exhaustive; designing compression methods explicitly optimized for fairness remains important future work.
> >
> >
> > [2] Strubell et al., Understanding the Effect of Model Compression on Social Bias in Large Language Models
>
> **Q4. Is LayerNorm monototically increasing?**
> > A4. That is a great catch. It is a typo and has been corrected in the revision.

---

> > ### Comment · Reviewer_DJxS · 2025-11-23
> > **reply to authors**
> >
> > Thank you for the additional experiments and explanation. The authors have answered most of my concerns and have added more details/experiments in response to other reviewers. WINA is intuitive and simple to implement, and the experiments shows clear performance gains compared to the baselines --- I have adjusted my final rating accordingly.

---

> > > ### Author Response · Authors · 2025-11-23
> > > **Thank you**
> > >
> > > Thank you again for your time and consideration. We sincerely appreciate your thoughtful review and the constructive feedback, which has helped strengthen this work.

---

### Official Review · Reviewer_SJQR · 2025-11-03

**Soundness:** 3
**Presentation:** 3
**Contribution:** 3
**Rating:** 6
**Confidence:** 3

**Summary:**

The paper proposes WINA (Weight Informed Neuron Activation), a training-free sparse activation method that combines hidden state magnitudes with the weight matrix structure to guide neuron selection. WINA is proven to minimize approximation error under column-wise orthogonality and monotonic activation assumptions. In the experiments, it outperforms other training-free methods like CATS, R-Sparse and TEAL across multiple LLM architectures (Llama2/3, Mistral, Phi-4) and benchmarks (MMLU, GSM8K, HumanEval), achieving over 60% FLOPs reduction at 65% sparsity while preserving accuracy.

**Strengths:**

- The proposed method introduces a simple yet effective training-free sparse activation mechanism that combines both hidden-state magnitudes and the column-wise L2-norm of weight matrices to guide neuron selection.
- The theoretical analysis is rigorous and well structured, providing provably optimal approximation error bounds under clear and interpretable assumptions (column-wise orthogonality and monotonic activation).
- The experiments are comprehensive, covering multiple model architectures, quantization methods, and ablations, demonstrating consistent improvements that align with theoretical predictions.

**Weaknesses:**

- The models in the experiments are small dense LLMs. Large-scale or MoE architectures (e.g., DeepSeek-V3, Llama4, GPT-OSS) which are more common in product deployment workloads are not tested. It’s unclear whether WINA’s activation gating would maintain efficiency with expert routing sparsity in these larger models.
- The evaluation focuses on theoretical FLOPs reduction but lacks real-world inference measurements such as latency or throughput on inference frameworks. Without kernel-level or runtime validation, the practical performance benefits of WINA remain unclear, especially given the hardware inefficiency of non-structured sparsity.
- The theoretical assumptions rely on column-wise orthogonality and monotonic activation functions, which may not strictly satisfied in real transformer models.

**Questions:**

- How does WINA perform on large MoE models (e.g., DeepSeek, Llama4, GPT-OSS)? It would help to understand how the method scales to production scale LLMs.
- Could you evaluate WINA’s actual latency or throughput in real inference scenarios? What’s the challenges to integrate this method to inference frameworks?
- How does WINA perform under long-context settings (e.g., 16K–128K tokens)? Are the top-K activation patterns stable as sequence length increases?

---

> ### Author Response · Authors · 2025-11-20
> **Response to reviewer SJQR**
>
> We appreciate Reviewer SJQR's constructive review and comments. Below, we provided point-to-point responses in line with the reviewer's questions and comments.
>
> **Q1. How does WINA perform on large MoE models (e.g., DeepSeek, Llama4, GPT-OSS)?**
> > A1. Thank you for the thoughtful suggestion. Extending WINA to large Mixture-of-Experts (MoE) architectures is indeed a promising direction. Conceptually, WINA provides fine-grained, in-layer activation gating (selecting which columns/rows to activate), whereas standard MoE routing performs coarse-grained expert selection (selecting which expert layers to activate). Combining these two mechanisms could lead to a **hybrid MoE system** that enables sparsification at multiple granularities and potentially offers additional efficiency gains at production scale.
> >
> > Due to the engineering complexity of such large-scale MoE systems and the limited bandwidth during the rebuttal period, we might be unable to include experiments on DeepSeek, Llama-4, or GPT-OSS at this time. However, this is an exciting direction, and we plan to explore hybrid MoE–WINA integration in future work as this question raises several methodological and architectural considerations that warrant a separate, in-depth investigation on its own.
>
> **Q2. Could you evaluate WINA’s actual latency or throughput in real inference scenarios?**
> > A2. Thanks for the important question. We have developed a custom WINA kernel using Triton. Then we tested it over varying GPU devices (A800, A100, RTX PRO 6000 Blackwell), varying batch sizes (1-256), and matrix dimensions. A brief numerical result is provided in the general response Q1/A1. More details realistic wall-time results could be found at Section 4.4, A.4, and A.5 in the revision. Overall, under the same sparsity level, WINA achieves nearly identical wall-clock performance to TEAL, which aligns with expectations since WINA’s additional computations are either pre-computed during model loading or negligibly lightweight. Combining with the noticeably higher performance of WINA at the same sparsity level, WINA thus achieves better efficiency and performance trade-off.
>
> **Q3. What’s the challenges to integrate this method to inference frameworks?**
> > A3. Thank you for the question. In our implementation of the WINA kernel using Triton, we did not encounter challenges beyond those already present in prior sparse-activation methods. This is expected, as WINA additionally requires two lightweight operations:
> > - a pre-computation of column/row weight norms, and
> > - an elementwise gating multiplication during inference.
> >
> > Both operations are fully supported by standard inference-framework primitives. As a result, WINA can be integrated into existing inference frameworks with no additional structural changes compared to prior sparse-activation kernels.

---

> ### Author Response · Authors · 2025-11-20
> **Continued response to reviewer SJQR**
>
> **Q4. How does WINA perform under long-context settings (e.g., 16K–128K tokens)? Are the top-K activation patterns stable as sequence length increases?**
> > A4. Thanks for the great suggestion. To the best of our knowledge, long-context setting is indeed not evaluated in the past sparse-activation domain. To provide the insights, we select the popular LongBench benchmark [1], which includes realistic long-context tasks such as code completion, summarization, and single and multi-document QA. LongBench comprises 4,750 examples with average input lengths ranging from 5K to 15K tokens.  Our evaluation covers Llama-2-7B, Llama-3-8B, Mistral-7B, and Phi-4-14B, which support context windows of 4K, 8K, 32K, and 16K tokens, respectively. Across nearly all settings, WINA consistently outperforms TEAL overall, including in more demanding long-context scenarios. In the meantime, the top-K patterns are quite stable (the same K leads to improved performance overall compared to TEAL across sequence lengths). These results are reflected in our updated manuscript of which we include a subset below.
> >
> > [1] LongBench: A bilingual, multitask benchmark for long context under-
> standing.
>
> >**Results of Llama-2-7B on LongBench**
> >
> >| Sparsity | Method | Code Completion | Few-shot Learning | Summarization | Multi-Document QA | Single-Document QA | Synthetic Tasks | Overall |
> >|--|--|--|--|--|--|--|--|--|
> >| 0 | Benchmark | 23.56 | 13.80 | 16.10 | 60.36 | 12.76 | 9.87 | 23.14 |
> >| 0.25 | TEAL | 62.14 | 51.80 | 7.24 | 12.87 | 11.83 | 5.06 | 22.59 |
> >| | **WINA**| 62.54 | 52.46 | 12.95 | 12.48 | 5.26 | 7.07 | **22.89** |
> >| 0.4 | TEAL | 60.28 | 52.23 | 8.12 | 13.40 | 12.55 | 4.54 | 22.83 |
> >| | **WINA**| 61.52 | 51.76 | 7.63 | 13.64 | 13.81 | 4.97 | **23.11** |
> >| 0.5 | TEAL | 58.41 | 50.10 | 8.39 | 12.80 | 12.97 | 7.09 | 22.23 |
> >| | **WINA**| 61.15 | 52.16 | 7.41 | 12.09 | 14.62 | 3.13 | **22.71** |
> >| 0.65 | TEAL | 44.20 | 8.17 | 1.62 | 11.16 | 14.15 | 7.88 | 17.88 |
> >| | **WINA**| 55.03 | 48.93 | 6.42 | 8.60 | 2.85 | 8.98 | **19.54** |
> >
> >Results of Llama-3-8B on LongBench
> >
> >| Sparsity | Method | Code Completion | Few-shot Learning | Summarization | Multi-Document QA | Single-Document QA | Synthetic Tasks | Overall |
> >|--|--|--|--|--|--|--|--|--|
> >| 0 | Benchmark | 23.56 | 13.80 | 16.10 | 60.36 | 12.76 | 9.87 | 23.14 |
> >| 0.25 | TEAL | 22.00 | 13.92 | 15.77 | 60.17 | 11.03 | 9.96 | 22.69 |
> >| | **WINA**| 23.30 | 13.65 | 16.24 | 60.20 | 10.92 | 9.89 | **22.82** |
> >| 0.4 | TEAL | 21.06 | 13.71 | 15.63 | 60.46 | 4.74 | 9.55 | 21.61 |
> >| | **WINA**| 24.13 | 13.02 | 17.71 | 60.47 | 7.30 | 9.74 | **22.57** |
> >| 0.5 | TEAL | 17.17 | 15.92 | 4.41 | 60.50 | 14.22 | 8.29 | 21.28 |
> >| | **WINA**| 23.88 | 19.29 | 3.95 | 60.24 | 11.82 | 7.73 | **21.97** |
> >| 0.65 | TEAL | 7.85 | 13.86 | 3.48 | 51.25 | 17.05 | 9.78 | 17.05 |
> >| | **WINA**| 19.58 | 3.07 | 7.07 | 53.61 | 14.66 | 7.07 | **18.13** |
> >
> >Results of Mistral-7B on LongBench
> >
> >| Sparsity | Method     | Code Completion | Few-shot Learning | Summarization | Multi-Document QA | Single-Document QA | Synthetic Tasks | Overall |
> >|--|--|--|--|--|--|--|--|--|
> >| 0        | Baseline| 65.74 | 51.51 | 11.82 | 8.66 | 19.21 | 5.75 | 24.45 |
> >| 0.25     | TEAL| 65.45 | 53.25 | 11.31 | 8.85 | 18.35 | 6.50 | 24.64 |
> >|          | **WINA**| 65.95 | 52.27 | 11.45 | 8.89 | 19.35 | 6.35 | **24.70** |
> >| 0.4      | TEAL| 64.70 | 53.44 | 9.84  | 8.94 | 17.96 | 5.88 | 24.18 |
> >|          | **WINA**| 65.41 | 53.15 | 9.88  | 9.00 | 17.83 | 6.13 | **24.22** |
> >| 0.5      | TEAL| 64.15 | 53.32 | 7.57  | 7.99 | 15.66 | 4.91 | 22.91 |
> >|          | **WINA**| 64.52 | 51.39 | 7.80  | 9.09 | 16.85 | 4.38 | **22.98** |
> >| 0.65     | TEAL| 56.70 | 52.36 | 6.22  | 7.45 | 14.00 | 0.52 | 20.72 |
> >|          | **WINA**| 59.09 | 48.74 | 9.71  | 7.73 | 11.83 | 3.34 | **20.96** |
> >
> >Results of Phi-4-14B on LongBench
> >
> >| Sparsity | Method | Code Completion | Few-shot Learning | Summarization | Multi-Document QA | Single-Document QA | Synthetic Tasks | Overall |
> >|--|--|--|--|--|--|--|--|--|
> >|0|Baseline | 29.58 | 56.03 | 8.41 | 4.80 | 18.59 | 59.59 | 28.06 |
> >|0.25|TEAL| 31.16 | 54.83 | 10.11 | 5.73 | 18.32 | 55.60 | 27.86 |
> >||**WINA** | 30.15 | 55.53 | 9.07  | 6.48 | 19.34 | 57.46 | **28.30** |
> >|0.4|TEAL| 30.65 | 55.20 | 12.53 | 9.95 | 21.38 | 48.70 | 28.74 |
> >||**WINA**| 33.51 | 56.13 | 11.68 | 11.28 | 19.60 | 52.10 | **29.43** |
> >|0.5|TEAL| 29.51 | 58.44 | 15.04 | 11.69 | 22.15 | 51.03 | 30.54 |
> >||**WINA**| 36.53 | 59.39 | 13.39 | 15.98 | 21.19 | 48.76 | **31.39** |
> >|0.65|TEAL| 25.71 | 59.48 | 14.11 | 19.25 | 19.50 | 43.52 | 30.07 |
> >||**WINA**| 37.60 | 59.22 | 17.36 | 12.24 | 16.30 | 46.58 | **30.26** |

---

> > ### Comment · Reviewer_SJQR · 2025-11-25
> > **Reply to authors**
> >
> > Thanks for your detailed explanation and response to my questions! Most of my concerns are answered. The additional benchmark result on LongBench also looks good and convincing. The final rating is adjusted.

---

> > > ### Author Response · Authors · 2025-11-25
> > >
> > > Thank you again for your time and your feedback. We very much appreciate the opportunity to improve our work based on your thoughts and review.

---

### Official Review · Reviewer_zwra · 2025-11-04

**Soundness:** 3
**Presentation:** 3
**Contribution:** 3
**Rating:** 6
**Confidence:** 2

**Summary:**

The paper addresses the important challenge of reducing inference cost in LLMs without degrading output quality. Existing training-free sparse activation methods often rely solely on hidden state magnitudes, which can lead to significant approximation error, particularly at high sparsity levels.

The authors propose WINA (Weight-Informed Neuron Activation), a simple yet effective framework that incorporates both hidden state magnitudes and the ℓ2-norm of weight matrices into neuron selection. This provides a principled sparsification strategy with provably optimal approximation error bounds, yielding tighter theoretical guarantees than prior methods.

The method is empirically validated across multiple widely used LLMs, including Llama-2-7B, Llama-3-8B, Mistral-7B, and Phi-4-14B, and evaluated on diverse tasks such as general reasoning (MMLU), mathematics (GSM8K), and coding (HumanEval). WINA is compared against several strong baselines, including TEAL, R-Sparse, and CATS. The results show that WINA performs comparably to prior methods at low sparsity and significantly better at high sparsity, achieving several percent improvement in commonsense reasoning accuracy and sustaining performance under extreme sparsity levels.

Overall, WINA is presented as a practical, theoretically grounded, and broadly deployable approach for efficient inference in LLMs.

**Strengths:**

The problem addressed is highly relevant, as reducing LLM inference cost without sacrificing output quality is an important challenge. The method is theoretically grounded, as incorporating weight norms provides a principled sparsification strategy with provable error bounds. The empirical evaluation is extensive, covering multiple LLMs, a range of tasks, and both low and high sparsity levels, and the method is compared to strong baselines including TEAL, R-Sparse, and CATS. The approach is practical and easy to deploy, as it is training-free and plug-and-play, making it broadly applicable. The results demonstrate robustness, as the method maintains competitive performance across different sparsity regimes and models.

**Weaknesses:**

The main contribution is a relatively straightforward extension of existing sparse activation methods, which could be considered incremental, though it is strengthened by solid theoretical and empirical support. The paper could benefit from a discussion of potential limitations, such as scenarios where weight-informed selection might be less effective or challenges when scaling to very large models beyond those tested.

**Questions:**

The authors propose incorporating weight norms into the neuron selection process. Could the authors clarify whether this additional step increases computation amount during inference, and if so, provide benchmark comparisons to quantify the overhead relative to other training-free sparse activation methods?

---

> ### Author Response · Authors · 2025-11-20
> **Response to reviewer zwra**
>
> We appreciate the constructive review and comments. Below, we provide our detailed answers in line with your questions and suggestions.
>
> **Q1. Add a limitation discussion.**
> > A1. Thanks for the suggestion. We have included a discussion on limitations in Appendix A.10 of our revised manuscript that mentions more explicitly the need to pre-compute the weight norm and orthogonality transformation to maximize WINA's performance.
>
> **Q2. Computational overhead of weight norm and quantifying that.**
> > A2. Thanks for the important question. The computational overhead of weight norm is negligible. In particular,
> > - The column weight norm $c_j=\|W_{:.j}\|_2$ can be pre-computed during model loading with complexity $O(d^2)$. And this pre-computational cost is amortized over all inference tokens.
> > - During inference, WINA adds only a single vector-wise multiply with complexity $O(BTd)$ which is typically $<0.1$% compared with  $O(BTd^2)$ due to the large $d$, the complexity of linear layer in modern LLMs.
> > - We developed a WINA kernel using Triton. In realistic throughput measurements across multiple GPU types (A800, A100, RTX PRO 6000), WINA achieves almost identical wall-time compared to TEAL while providing larger accuracy gains at high sparsity. Detailed results are present at Section 4.4, and Appendix A.4 and A.11 in the revised version of our manuscript.
> >
> > Please refer to our general response for more details, additional info, and clearer notation.

---

### Official Review · Reviewer_i21q · 2025-11-17

**Soundness:** 2
**Presentation:** 2
**Contribution:** 1
**Rating:** 2
**Confidence:** 3

**Summary:**

- Proposes WINA, a training-free sparse activation method that gates neurons using both activation magnitude and column-wise weight ℓ₂ norms.

**Strengths:**

- Very simple, plug-and-play rule that is easy to implement on top of existing sparse-activation baselines.

**Weaknesses:**

- The paper reports GFLOP reductions but does not clearly explain whether WINA’s gating is used to avoid weight loads or only to mask post-matmul activations; without a truly sparse kernel and latency measurements, it is unclear how much real speedup WINA provides over TEAL/CATS in memory-bound, batch-1 inference.

**Questions:**

- In your current implementation, is the WINA gate used before the matmul to index only a subset of columns of 𝑊 , or do you compute a dense 𝑊𝑥 and then apply the mask? If it’s the latter, how do you obtain any wall-clock speedup, especially in batch-1, memory-bound settings?

- Comparison to TEAL/CATS kernels: TEAL/CATS explicitly discuss sparse kernels that reduce weight loading per token. Do you implement a comparable kernel for WINA, and can you report latency/throughput numbers vs TEAL/CATS on real hardware (A100, L40S, etc.), not just GFLOP estimates?

---

> ### Author Response · Authors · 2025-11-20
> **Response to reviewer i21q**
>
> We thank the reviewer for the constructive comments and for recognizing the simplicity and plug-and-play nature of WINA. Below we address all concerns in detail. We hope our clarified explanations and newly added experiments address the reviewer's questions about real speedup and kernel implementation.
>
>
> **Q1. In your current implementation, is the WINA gate used before the matmul to index only a subset of columns of 𝑊 , or do you compute a dense 𝑊𝑥 and then apply the mask? If it’s the latter, how do you obtain any wall-clock speedup, especially in batch-1, memory-bound settings?**
> > A1. Thanks for the thoughtful question. Algorithmically, WINA apply its gating before the matmul to directly index the active subset of columns of $W$. In practice, we provide two implementations, one is implemented using torch, one is implemented using Triton. In both versions, we strictly follow the method and do not compute a dense $Wx$ then apply a mask.
> >
> > In particular,
> > - During model loading, the column-wise weight norm $c$ is computed once and cached, introducing no runtime overhead during inference.
> > - During inference, $c$ is multiplied elementwise with the activation vector $x$ to generate the WINA gate $g$ at the desired sparsity level. This gate directly selects a subset of columns of $W$, which are the only columns loaded and multiplied inside the Triton kernel.
>
>
>
> **Q2. Do you implement a comparable kernel for WINA, and can you report latency/throughput numbers vs TEAL/CATS on real hardware (A100, L40S, etc.), not just GFLOP estimates?**
> > A2. Thanks for the important question. Yes, we have implemented a dedicated WINA Triton kernel, and it has been extensively tested across multiple GPU devices (A100, A800 and RTX-6000 Blackwell), batch sizes (1-256), and matrix dimensions. The results show that WINA achieves nearly identical acceleration and wall-time speed-up as TEAL when compared against the dense baseline, confirming that WINA provides practical efficiency. Detailed latency curves are included in General Response Q1/A1, Section 4.4, and Appendix A.4 and A.11 in the revised version. Importantly, because WINA delivers noticeably higher model performance under the same sparsity level, it consequently offers higher acceleration at the same accuracy level than prior state-of-the-art methods such as TEAL.
>
> We hope these clarifications and the newly added comprehensive Triton kernel experiments fully resolve the concerns regarding realistic speedup. We would be grateful if you could kindly reconsider the overall assessment and increase the rating if appropriate.

---

> ### Author Response · Authors · 2025-11-26
> **Gentle reminder**
>
> Dear Reviewer i21q,
>
> Thank you again for your thoughtful feedback. Your questions about the realistic speedup (which is a question shared by other reviewers) and whether WINA performs gating before the matmul were helpful in strengthening our work.
>
> Following your questions/suggestions, we have added:
>
> - **A fully implemented WINA Triton kernel.**
>
> - Realistic wall-clock latency and throughput measurements across different GPUs (A100/A800/RTX6000 Blackwell), sparsity levels, and batch sizes. The results show that **under the same sparsity level**, WINA achieves **almost identical speed-up** to TEAL. Consequently, **at the same accuracy level**, WINA achieves **higher speed-up** than TEAL.
>
> - A detailed explanation showing that **WINA always indexes and selects columns before matmul** to achieve realistic speedups.
>
> Detailed content can be found in Section 4.4 and the Appendix of our revised manuscript; these new additions/results have been well-received by other reviewers with similar concerns before rebuttal.
>
> Since the discussion period deadline is approaching within a week, we kindly invite the reviewer to consider whether the newly added content and experiments have addressed concerns and to re-assess our work if appropriate. We are also very happy to provide any further clarification that might be helpful. Thank you again for your time and consideration.
>
> Sincerely,
>
> Authors

---

### Author Response · Authors · 2025-11-17
**General Response**

We sincerely appreciate all reviewers and ACs for the constructive reviews and insightful comments. We are glad to see the broad agreement that:

- The method is simple, plug-and-play, and practically useful, (reviewer `i21q`, `zwra`, `SJQR`, `DJxS`, `1iPj`, `iVmy`).
- The theoretical analysis is solid and improves over prior work, (reviewer `zwra`, `SJQR`, `DJxS`, `iVmy`).
- The empirical coverage is extensive, demonstrates clear advantages at varying sparsities and is robust across architectures, tasks, and even quantized precision, (reviewer `zwra`, `SJQR`, `DJxS`, `1iPj`, `6SUJ`).

Below we address two common questions shared across reviewers. Responses to the remaining questions and the revision of the paper will be **provided in the next couple of days**.

**Q1. WINA's performance over realistic wall-time or throughput.**
> A1. We appreciate this important question. We have now implemented a **Triton-based sparse-activation kernel** and measured real inference throughput on varying size of matrix-vector multiplications. Here are two takeaways.
> - **Negligible FLOPs complexity overhead.** Weight norms $c_j=||W_{:,j}||_2$​ are **pre-computed once offline** during model loading. During inference the gating requires only a vector-wise elementwise multiplication $x \odot c$ compared to other approaches, e.g., TEAL and CATS. Both operations are fully supported by standard inference-framework with no additional structural changes.
>
>   Regarding complexity, let $d$ be the hidden dimension, $B$ the batch size, and $T$ the sequence length.
>     - A standard linear layer forward consumes $O(BTd^2)$ FLOPs.
>     - The WINA gating mechanism requires $O(BTd)$.
>
>   Comparing orders, $\frac{O(BTd)}{O(BTd^2)}=O(\frac{1}{d})$. In modern LLMs, $d$ are typically large numbers, e.g, 2048, 4096. As a result, WINA consumes only $<0.1$% additional overhead, which is negligible especially compared with reduced FLOPs and additional performance gain.
> - **Realistic Throughput Speedup.** With our developed **Triton kernel**, WINA delivers **measurable realistic speedup**. Early results showed that under A800 GPU, WINA achieves **almost identical realistic speedups** to TEAL. We are conducting more experiments over other GPUs, e.g., A100, and will provide soon.
>
> GPU: A800, Wall-Time Unit: millisecond, Repeating Times: 10
>
> Matrix: 4096x11008
>|Sparsity | WINA| TEAL| Theoretical Optimal|
>|--|--|--|--|
>| 0 | 0.0747 | 0.0747 | 0.0747 |
>| 25% | 0.0583 | 0.0593 | 0.0560 |
>| 40% | 0.0511 | 0.0511 | 0.0448 |
>| 50% | 0.0471 | 0.0471 | 0.0373 |
>| 65% | 0.0419 | 0.0409 | 0.0261 |
>
>  Matrix: 4096x14336
> |Sparsity | WINA| TEAL| Theoretical Optimal|
> |--|--|--|--|
> | 0 | 0.0972 | 0.0972 | 0.0972 |
> | 25% | 0.0706 | 0.0706 | 0.0737 |
> | 40% | 0.0624 | 0.0614 | 0.0589 |
> | 50% | 0.0573 | 0.0552 | 0.0491 |
> | 65% | 0.0501 | 0.0491 | 0.0344 |
>
> Matrix: 5120x17920
> |Sparsity | WINA| TEAL| Theoretical Optimal|
> |--|--|--|--|
> | 0 | 0.1290 | 0.1290 | 0.1290 |
> | 25% | 0.1003 | 0.1013 | 0.0967 |
> | 40% | 0.0870 | 0.0860 | 0.0774 |
> | 50% | 0.0798 | 0.0778 | 0.0645 |
> | 65% | 0.0706 | 0.0686 | 0.0451 |


**Q2. How to enforce the column-orthogonality holds in general practice.**
> A2. We appreciate this insightful question. In WINA, our contribution is the introduction of weighted-information gating mechanisms, while not the orthogonality transformation, which is used for the theoretical optimality proofs as Lemma 3.1, to decouple interactions so that the approximation-error decomposition becomes analytically tractable (cross terms vanish).
> To enforce or approximate column-orthogonality on DNNs, there are two general families of techniques.
> - One is to add orthogonality regularizer $||W^\top W-I||$ into the loss function, representative works includes
>
>    [1] All You Need is Beyond a Good Init: Exploring Better Solution for Training Extremely Deep Convolutional Neural Networks with Orthonormality and Modulation
>
>    [2] Controllable Orthogonalization in Training DNNs.
> - The other is to use matrix decomposition and transformation to enforce the column-orthogonality. We adopted the method proposed in SliceGPT [3]. As described in the remark Section 3.3, this transformation is conducted one-shot beforehand and resulting in a numerical equivalent (computational invariant) LLMs for further usages.
>
>    [3] SliceGPT: Compress Large Language Models by Deleting Rows and Columns.
>
>    This transformation is also widely adopted in efficient-model literature, including [4-7].
>
>    [4] QuaRot: Outlier-Free 4-Bit Inference in Rotated LLMs.
>
>    [5] QuiP: 2-Bit Quantization of Large Language Models with Guarantees.
>
>    [6] SpinQuant: LLM Quantization with learned rotations.
>
>    [7] OstQuant: Refining Large Language Model Quantization with Orthogonal and Scaling Transformations for Better Distribution Fitting.

Thanks all again. Look forward to further discussions.

Yours,

Authors

---

### Author Response · Authors · 2025-11-20
**Additional General Response**

We thank all the reviewers again for their insightful feedback which has improved the quality of our paper. We provide a general response here along with individual point-to-point responses to each reviewer further below. We have also uploaded a revised version of our manuscript reflecting changes and additional experiments/results wherein updated portions of the manuscript are marked in $\color{blue}{\text{blue}}$.

Below, we highlight three main groups of newly added experiments suggested during the rebuttal process. The latter two present newly important aspects which are typically not shown in existing sparse activation literature (i.e., effect on social bias and effect on long-context reasoning ability).

- **Experiment 1. Realistic Speed-up via WINA Triton Kernel.**
    -  Per the suggestion of multiple reviewers, we have implemented a custom WINA Triton kernel to benchmark realistic latency and speed-up performance.
    - We have tested it over different GPUs (A800, A100, RTX PRO 6000 Blackwell), matrix-vector multiply sizes, and batch sizes.
    - The updated results are provided below and in our revised manuscript (Section 4.4 and Appendix A.10, A. 11) and Q1/A1 in our General Response.
    - Overall, at the same sparsity level, WINA achieves almost identical realistic speed-ups to TEAL. Consequently, at the same accuracy level, WINA gating mechanism attains higher speed-up than TEAL, highlighting the advantage of WINA’s accuracy–efficiency trade-off.

- **Experiment 2. Sparse Activation on Social Bias.**
	- To our knowledge, existing sparse activation works do not document the effects of their respective sparse activation methods on model/social bias.
	- As suggested by reviewer `DJxS`, we have conducted experiments following [1] where we evaluate our WINA-sparsified LLMs on CrowS-Pairs across Gender, Race, and Religion. These results are present in our updated manuscript (Appendix A.6).
	- Across all architectures and sparsity levels, we observe no systematic increase in bias, instead sparse activated models often move closer to the optimal 50% neutrality target, which is consistent as the compressed models in [1]. These results suggest that WINA does not exacerbate bias and can modestly mitigate it.

	[1]. Strubell et al., Understanding the Effect of Model Compression on Social Bias in Large Language Models

- **Experiment 3. Sparse Activation on Long-context Benchmark, LongBench.**
	- Similarly, to the best of our knowledge, assessing how well sparse activated models perform in long-context reasoning settings is typically not evaluated in this domain.
	- As suggested by reviewer `SJQR`, to provide insight into WINA’s effects, we adopt the widely-used LongBench benchmark to assess model performance across different sparsity levels. LongBench covers six long-context scenarios across 4,750 examples. We report results for Llama-2-7B, Llama-3-8B, Mistral-7B, and Phi-4-14B.
	- The results have been updated in our revised manuscript (Section 4.5 and Appendix A.5). In general, we can see that across all settings overall, WINA consistently outperforms TEAL in long-context scenarios.

We kindly invite reviewers to assess whether our revisions and additional experiments address their concerns. We would be grateful if ratings can be updated where appropriate. We thank the reviewers for their time, constructive feedback, and thoughtful engagement with our work.

Sincerely,

Authors

---

### Author Response · Authors · 2025-11-29
**Summary of rebuttal process so-far**

Dear ACs and Reviewers,

Thank you very much for your valuable time, constructive feedback, and active engagement throughout the review process. We were all surprised by the recent circumstances on Nov 27, 2025 and we appreciate the decision to revert to the pre-discussion state as an important step to protect the fairness and integrity of our community.

To assist with evaluation, we provide a concise summary of the review and discussion process thus far, along with the updates we have made during the rebuttal process.

**Paper Summary**

This work proposes **WINA**, a novel but simple training-free sparse activation method that significantly accelerates LLM inference. We provide:

- **Simple and effective training-free LLM inference speedup**: WINA introduces a novel weight-informative aware activation to significantly accelerate LLM inference with stronger performance preservation.
- **Strong theoretical backing**: We establish rigorous bounds showing tighter approximation error of WINA than previous approaches.
- **Comprehensive experimentation**: WINA achieves considerable gains consistently across various model architectures, benchmarks (including two new ones added during rebuttal addressing WINA’s impact on social/model bias and long-context), and quantization/precisions (FP16/INT8/INT4).
- **Real-world efficiency**: WINA produces superior speedup compared to SOTA methods such as TEAL at the matched accuracy level with a custom kernel implementation for real world latency measurements (added during rebuttal).

Across all settings (theoretically and empirically) overall, WINA consistently presents clear advantages over other SOTA methods.

**Review Status Before Author Response**

Before the discussion period began on Nov 11, 2025, we received six reviews (scores: `6,6,6,6,6,4`). On Nov 17, 2025 (5-6 days after the discussion period had begun), an additional review from reviewer `i21q` (score `2`) was posted.

Across reviewers, the consensus is:

-   **Simplicity and usefulness**: the method is simple, plug-and-play, and practically valuable (reviewers `i21q`, `zwra`, `SJQR`, `DJxS`, `1iPj`, `iVmy`).
-   **Theoretical rigor**: the analysis is solid and improves upon prior work (reviewers `zwra`, `SJQR`, `DJxS`, `iVmy`).
-   **Comprehensive experiments**: the empirical study is extensive, shows clear advantages at varying sparsities, and is robust across architectures, tasks, quantized precision, and **realistic speedups (after responses)** (reviewers `zwra`, `SJQR`, `DJxS`, `1iPj`, `6SUJ`).

The **main review question** centered on whether we implemented **a dedicated Triton kernel for WINA and how its realistic speedup compares to SOTA sparse-activation methods such as TEAL**.

**Review Status After Author Response**

In response, we have addressed the main concern by implementing a **WINA-specific Triton kernel** and conducted extensive real-world throughput experiments. These results show that across GPUs, batch sizes, and matrix dimensions, WINA achieves **nearly identical speedups to TEAL at equal sparsity** and **higher speedups at matched accuracy**. We have attached the full source code for reproducibility.

Following these updates, multiple reviewers confirmed that their concerns had been resolved. Consequently:

-   Reviewer `DJxS` increased the rating from `6 → 8` (Nov 23, 2025) after confirming the issues were addressed.
-   Reviewer `SJQR` increased the rating from `6 → 8` (Nov 25, 2025), noting concerns were resolved and the long-context experiments provided valuable new insights.
-   Reviewer `6SUJ` increased the rating from `4 → 6` (Nov 25, 2025) after finding the revisions satisfactory.
-   Reviewer `1iPj` acknowledged our clarifications and maintained the rating of `6 → 6` (Nov 25, 2025).

We appreciate that the majority of the reviewers have responded positively during the rebuttal process. Although reviewer `i21q`, the only negative rating, did not engage with our additional updates and results, we understand that might be due to the platform restricting reviewers from commenting further under these special circumstances. In the meantime, all the concerns of reviewer `i21q` were also raised by other reviewers, while these have been resolved during the rebuttal stage, as reflected in the other reviewers’ own replies and score increases.

We kindly ask the ACs to take this context into account. If there is anything further we can provide, we would be very happy to assist. Thank you again for your time and thoughtful consideration. We greatly appreciate your valuable efforts under these challenging conditions!

Sincerely,

Authors

---

### Meta-Review · Area_Chair_KDRh · 2025-12-31

**Summary:**

The authors have effectively addressed the primary concern regarding real-world latency. The addition of the custom kernel and subsequent performance benchmarks has strengthened the paper significantly. Following the rebuttal, three reviewers increased their scores (from 6→8, 6→8, and 4→6), and one maintained a positive score of 6. The single remaining negative review (score 2) did not engage with the rebuttal updates; however, their concerns appear to have been addressed by the new data and experimental results. Given the strong theoretical backing, comprehensive empirical results, and successful response to the main reviewer's critique, this paper makes a valuable contribution to efficient LLM inference.

**Reviewer Concerns:**

The primary area of concern revolved around real-world implementation and efficiency. Specifically, reviewers questioned the availability of a dedicated custom kernel for WINA and how its realistic speedup compared to state-of-the-art methods like TEAL.

In response, the authors implemented a custom Triton kernel for WINA and conducted extensive real-world throughput experiments. The new results show that WINA achieves comparable speedups to TEAL at equal sparsity and superior speedups when matched for accuracy. The source code was also provided for reproducibility.

**Reviewer Scores:**

All reviewer could raise their scores.

---

### Decision · Program_Chairs · 2026-01-26

Accept (Poster)